# Human endogenous oxytocin and its neural correlates show adaptive responses to social touch based on recent social context

Linda Handlin[1†], Giovanni Novembre[2†], Helene Lindholm[1], Robin Kämpe[3,4], Elisabeth Paul[3,4], India Morrison[2,3]*

[1]Department of Biomedicine, School of Health Sciences, University of Skövde, Skövde, Sweden; [2]Division of Neurobiology, Department of Biomedical and Clinical Sciences, Linköping University, Linköping, Sweden; [3]Center for Medical Image Science and Visualization (CMIV) Linköping University Hospital, Linköping, Sweden; [4]Center for Social and Affective Neuroscience, Department of Biomedical and Clinical Sciences, Linköping University, Linköping, Sweden

*For correspondence: india.morrison@liu.se

[†]These authors contributed equally to this work

Competing interest: The authors declare that no competing interests exist.

**Abstract** Both oxytocin (OT) and touch are key mediators of social attachment. In rodents, tactile stimulation elicits the endogenous release of OT, potentially facilitating attachment and other forms of prosocial behavior, yet the relationship between endogenous OT and neural modulation remains unexplored in humans. Using a serial sampling of plasma hormone levels during functional neuroimaging across two successive social interactions, we show that contextual circumstances of social touch influence not only current hormonal and brain responses but also *later* responses. Namely, touch from a male to his female romantic partner enhanced her subsequent OT release for touch from an unfamiliar stranger, yet females' OT responses to partner touch were dampened following stranger touch. Hypothalamus and dorsal raphe activation reflected plasma OT changes during the initial social interaction. In the subsequent interaction, precuneus and parietal-temporal cortex pathways tracked time- and context-dependent variables in an OT-dependent manner. This OT-dependent cortical modulation included a region of the medial prefrontal cortex that also covaried with plasma cortisol, suggesting an influence on stress responses. These findings demonstrate that modulation between hormones and the brain during human social interactions can flexibly adapt to features of social context over time.

## Editor's evaluation

This fundamental work combined naturalistic and neuroscientific methods to demonstrate the context-dependent impact of oxytocin on the brain and behavior. The authors provide compelling evidence that adds significant nuance to our understanding of how social touch is mediated by the brain, which can render people both more and less trusting, depending on conditions. This work will be of broad interest to psychologists and neuroscientists at many levels.

## Introduction

A hug from a friend, a caress from a lover, the secure embrace of a parent: touch is a predominant channel for bolstering human connection and emotional attachment. The neural mechanisms supporting this vital role of touch are not fully understood, but there is evidence from rodents that

tactile stimulation in social interactions can act as a major trigger for the endogenous release of the neuropeptide hormone OT (*Kurosawa et al., 1995*; *Tang et al., 2020*), which has been implicated in the social attachment (*Feldman, 2012*; *Uvnäs-Moberg et al., 2014*). It has, therefore, been suggested that social tactile stimulation such as affectionate stroking may elicit endogenous OT release in adult humans (*Chen et al., 2020*; *Kreuder et al., 2017*; *Li et al., 2019*; *Walker et al., 2017*).

In this perspective, stimulation of tactile nerves in the skin initiates a cascade of modulatory responses in the brain, mediated by specific neural populations in the hypothalamus. It is well-established that magnocellular OT neurons in the paraventricular nucleus (PVN) of the hypothalamus send projections to the forebrain and cortical regions mediating olfactory and somatosensory signaling in the brain (*Burbach et al., 2005*; *Knobloch et al., 2012*; *Mitre et al., 2018*; *Oettl et al., 2016*). For tactile stimulation, this signaling may rely on a population of parvocellular OT neurons in the PVN selective for particular forms of affiliative touch (*Tang et al., 2020*; *Wang et al., 2022*; *Yu et al., 2022*). OT released into the bloodstream via a PVN-pituitary gland pathway modulates the action of vasculature and smooth muscle (*Qin et al., 2009*), notably during processes surrounding reproductive behavior in both sexes, as well as parturition and lactation in females (*Filippi et al., 2003*; *Uvnäs-Moberg, 1998*).

OT's broad role in parental nurturance and affiliative behavior may reflect a functional extension of its influence on these core reproductive and maternal behaviors (*Walum and Young, 2018*), many of which rely on sensory cues such as touch and olfaction, and can even encompass cross-species inter-actions (*Algoe et al., 2017*; *Nagasawa et al., 2015*; *Rehn et al., 2014*). Many OT-relevant sensory stimuli likely involve central pathways of OT in the brain, but peripheral mechanisms of release can also be triggered by stimulation of the genitals, the nipples, or the vagal nerve. Both central and peripheral mechanisms can play a role in such stimulus-driven effects of OT on social behavior (*Althammer et al., 2021*).

In contrast, the influence of exogenous, intranasal OT (IN-OT) administration on behavioral and neural outcomes has been widely studied with regard to social stimuli (*Di Simplicio et al., 2009*; *Hein-richs et al., 2003*) For example, IN-OT have shown varying effects on social outcome measures such as face processing (*Fischer-Shofty et al., 2010*; *Petrovic et al., 2008*; *Rimmele et al., 2009*), empathy (*Domes et al., 2007*), romantic relationships and bonding (*Ditzen et al., 2009*; *Kreuder et al., 2017*; *Scheele et al., 2012*; *Scheele et al., 2013*), and romantic touch (*Kreuder et al., 2019*). Neuroimaging studies of IN-OT manipulations indicate that many relevant changes occur at the cortical level (*Wang et al., 2017*; *Zink and Meyer-Lindenberg, 2012*), suggesting more complex modulatory pathways than is implied by a stimulus-driven model focused on signaling from afferent receptors to subcortical brain regions.

There is uncertainty surrounding the mechanisms of action of IN-OT and its degree of equivalence to endogenous release, chiefly regarding the questions of whether the molecule crosses the blood-brain barrier, and how peripheral effects can be disentangled from central effects (*Churchland and Winkielman, 2012*; *Striepens et al., 2013*; *Wang et al., 2017*; *Zink and Meyer-Lindenberg, 2012*, but see *McCullough et al., 2013*). It is, therefore, crucial to investigate endogenous OT changes and their neural correlates in humans for a fuller understanding of its relevant mechanisms and functional roles, as well as its potential limits and parameters.

One potential role for endogenous OT release in human social interactions may lie in stress regulation. OT modulates cardiovascular responses, playing a key role in regulating the autonomic nervous system (*Uvnäs-Moberg, 1998*). OT may also modulate stress responses via an interplay with hypothalamic-pituitary-adrenal (HPA) mechanisms affecting the hormone cortisol, a key mediator of stress responses. Research in rats has suggested a role in particular tactile-mediated social behaviors (*Tang et al., 2020*), while research in dogs has suggested a role for endogenous OT in touch during the social reunion with their owners, which may also involve cortisol changes (*Rehn et al., 2014*). In, humans, neuroimaging research on touch and OT has implicated the orbitofrontal cortex during foot massage (*Li et al., 2019*). No clear overall picture of a relationship between endogenous OT and the brain has yet emerged, however.

Social, affective touch has also been associated with a specific subtype of afferent nerve fiber called C-tactile (CT) afferents in humans (C-low-threshold mechanoreceptors or C-LTMRs in rodent models), which are found in hair-follicle-containing skin (*Löken et al., 2009*; *Morrison et al., 2010*; *Olausson et al., 2002*; *Walker et al., 2017*). CTs increase firing frequency for caress-like touch stimulation, which has correlated with subjective reports of touch pleasantness on a group level (*Walker et al.,*

*2017*). Because both OT and CTs have been implicated in affective touch, it has been proposed that OT may be involved in CT-related neural mechanisms (*Walker et al., 2017*). Any such link would be supported if brain regions preferentially responding to touch on the CT-rich skin of the arm, compared to the CT-poor skin of the palm, showed specific OT-brain covariation. Alternatively, any cortex-OT modulation for social touch may be independent of any particular peripheral nerve type. Further, a general OT-brain comodulation may depend more heavily on contextual factors such as the familiarity of the interacting individual, as well as aspects of recent experience and current physiological state.

In this study, we examined whether social interactions involving touch can evoke endogenous changes in plasma OT in human females and whether this would be modulated by interacting with a socially familiar individual. Given the preponderance of OT studies in male populations, we also took a 'female-first' strategy (*Shansky and Murphy, 2021*) by testing hormone and brain responses in a female population. This experiment investigated the influence of contextual variables on endogenous hormones and their relationship with brain changes by combining serial sampling of plasma OT and cortisol with functional magnetic resonance imaging (fMRI). We predicted that touch from a socially familiar person (a romantic partner) would evoke greater endogenous OT changes than touch from an unfamiliar person (a nonthreatening stranger), allowing investigation of the neural responses associated with any such modulation. On the neural level, we expected engagement of the hypothalamus and other key regions associated with evolutionarily-conserved circuitry of OT modulation and receptor expression in different species (such as the amygdala, medial prefrontal cortex, and cingulate cortex).

Crucially, we also explored whether these OT-brain modulation changes would be modulated by participants' very recent social interaction history with familiar or unfamiliar others. Each participant was caressed by both their partner and an unfamiliar yet unthreatening stranger over two successive parts of the same experimental session, while a total of eight plasma OT samples and six plasma cortisol samples per participant were collected over the session (*Figure 1A*). The presentation order of partner or stranger across the two successive touch interactions during the experiment was counterbalanced: the stranger's touch could either precede the partner's touch or come after it. If the order of partner/stranger presentation influences OT-brain modulation, this should be reflected in plasma OT and brain responses, though we did not hypothesize a direction for any such effect. The ultimate aim of this experimental design was to identify brain regions in which hemodynamic responses changed as a function of endogenous OT levels over the experimental session, thus revealing any context-sensitive, OT-dependent engagement of both subcortical, and cortical systems.

Beyond these basic questions regarding touch, familiarity, and recent social interaction history, we also investigated any relationship between brain-hormone modulation to stress responses by testing for covariation between OT and peripheral cortisol changes, which can be evoked during a mild, acute stress challenges such as interacting with a socially unfamiliar individual within the novel environment of an fMRI experiment. We predicted lower plasma cortisol levels for the partner compared to the stranger, and an inverse relationship between OT and cortisol measures, during social touch interactions. The paradigm also allowed for investigation of the question of whether brain regions showing differential responses to touch on arm skin, rich in CT afferent nerves, as compared to CT-poor palm skin, would show selective OT-dependent modulation.

## Results
### Behavioral measures
#### Self-report questionnaires
Before entering the fMRI scanner, both the participant and her partner separately filled out two questionnaires: the Couples Satisfaction Index (CSI; *Funk and Rogge, 2007*) and the State-Trait Anxiety Inventory (STAI; *Spielberger, 1970*). CSI scores indicated that both female and male participants were satisfied with their relationships (females: *M*=139.8, SD = 17.2, males: *M*=139.6, SD = 15.2), and the participants' assessments of relationship quality correlated with their partners' (*r*=0.57, p=0.0003). STAI scores indicated that no participants demonstrated clinically significant symptoms of anxiety (females STAI-S *M*=35.6, SD = 9.7; female STAI-T *M*=40.3, SD = 9.8; males STAI-S *M*=31.6, SD = 5.7; males STAI-T: *M*=36.2, SD = 6.7). The higher participants' trait anxiety, the lower their assessment of relationship quality, with an inverse correlation between STAI-T and CSI scores (*r*=–0.40, p=0.01 for

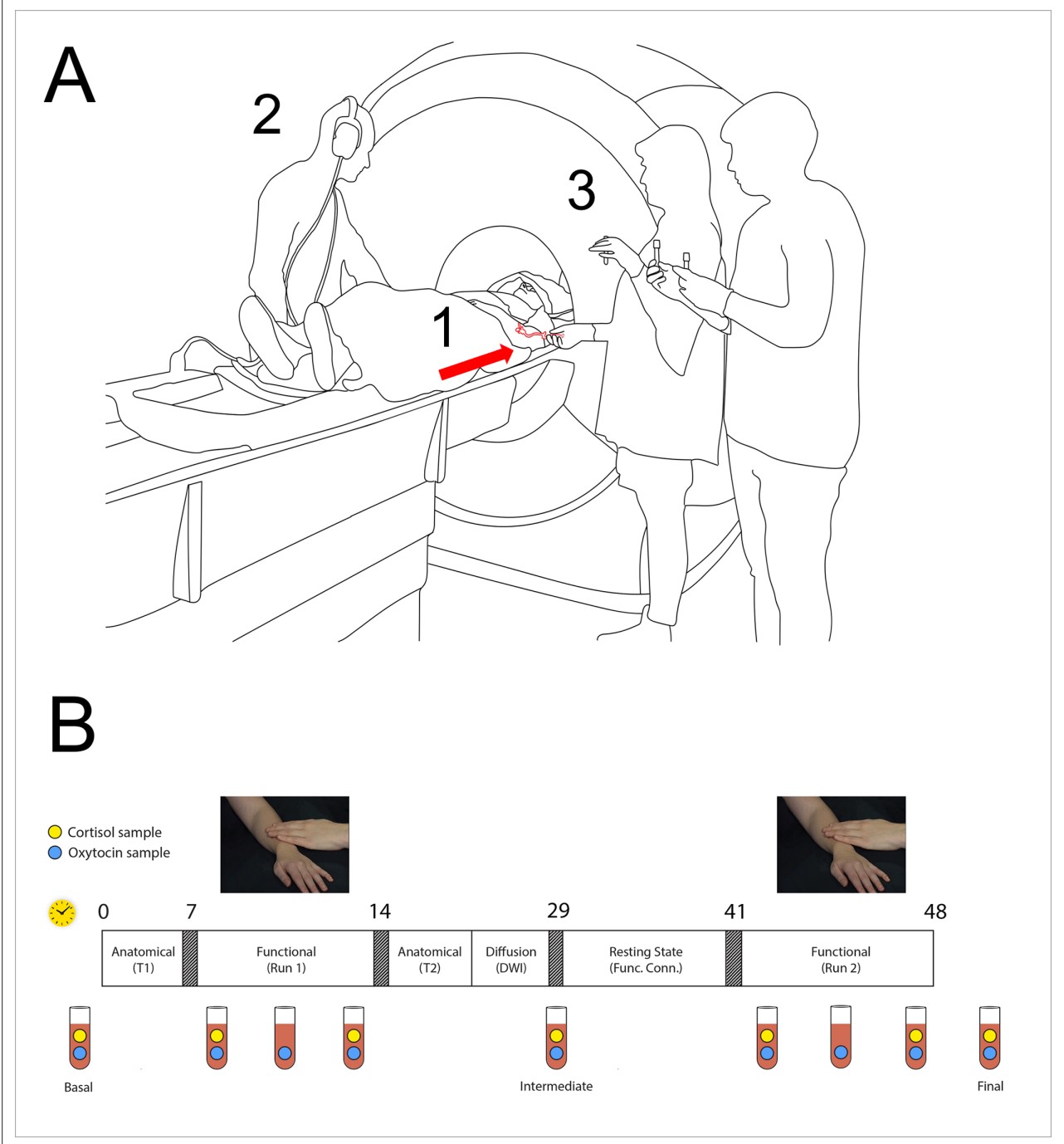

**Figure 1.** Functional magnetic resonance imaging (fMRI) experiment setup. (**A**) (1) Indwelling catheter in female participant's left arm (arrow); (2) participant's male partner or unfamiliar stranger caressed the participant, following audio prompts; (3) serial blood samples were collected from the catheter. (**B**) Structure of fMRI experimental session with a serial sampling of plasma oxytocin and cortisol. Rectangle depicts the time course of the experiment, with approximate elapsed minutes shown above (yellow clock symbol). Two functional runs with partner and stranger touch, in counterbalanced order, were separated by ~27 min. Three baseline oxytocin (OT) samples (1, 5, 9) and three serial samples were collected for each run (2-4, 6-8). Clock symbol indicates the time in minutes. Blue dots in the vial symbols indicate oxytocin samples, the green dots indicate cortisol samples. The first functional run was preceded by the acquisition of an anatomical (T1-weighted) image, while between functional runs additional anatomical and functional scans were acquired: T2-weighted anatomical image, diffusion-weighted imaged, and resting-state. See also *Materials and methods* below.

the participant, $r=-0.39$, p=0.01 for partner). There were no other significant correlations among CSI, STAI-T, STAI-S, and VAS pleasantness scores (all $ps$ <0.01).

## Ratings

For trial-by-trail ratings of touch pleasantness, partner touch was rated as more pleasant than stranger touch ($F_{(1, 33)}=30.032$, p<0.001) with a main effect of higher ratings for arm ($F_{(1, 33)}=11.070$, p=0.002, effect size $f=0.7$, *partial* $\eta^2=0.33$ at power (1-β error probability)=0.8, $\alpha=0.05$). Participants who received stranger touch first had lower ratings for stranger touch on the palm compared to stranger touch on the arm ($t=16$, p=0.007, $d=1.99$), reflected in a significant three-way interaction, indicating influences of both familiarity and order on touch pleasantness ratings for palm ($F_{(1, 33)}=4.730$, p=0.037).

For post-scanning ratings of trustworthiness and attractiveness, the stranger was rated as positively trustworthy, with a mean rating of 3.96 on a visual analog scale from –10 (untrustworthy) to 10 (trustworthy). Stranger attractiveness was near a neutral midpoint, with a mean rating of 0.95 on a visual analog scale from –10 (unattractive) to 10 (attractive). For participants starting with partner touch, evaluation of relaxation for partner touch correlated with how trustworthy participants rated the stranger ($r=0.77$, p=0.002). The interaction with the nurse was rated as non-stressful, with a mean rating of 5.28 on a scale of –10 (stressful) to 10 (calming). For participants in the stranger first group, the smaller the difference between pleasantness ratings for the stranger and partner touch during the imaging session, the more relaxing participants rated stranger touch afterward ($r=-0.70$, p=0.004). Within-session pleasantness ratings also predicted post-session evaluations of relaxation for partner and stranger touch independently (partner $r=0.83$, p=0.001 stranger $r=0.80$, p=0.001).

## Hormone analyses

### OT levels

A total of eight serial OT samples per person were collected during the course of the session (*Figure 1B*). As predicted, OT levels increased when the partner was the interactant in the first encounter (*Figure 2A*). In addition, when stranger touch was preceded by partner touch OT levels showed a significant dip and recovery during the second touch session. These results were revealed by a three-way interaction between the factors familiarity, order, and sample timepoint ($F_{(3, 183.180)}=3.034$, p=0.031). A significant increase between the first and middle samples in the functional run during stranger touch in the partner first group only (p=0.027) drove the contribution of timepoint to the three-way interaction. There was also a two-way interaction between the factors familiarity and order ($F_{(1, 183.169)}=11.216$, p=0.001). Marginally below-alpha differences were seen between the middle and final samples during the run, pre-run baseline, and the first sample during the first run in the partner first group (p=0.054 and p=0.070, respectively). There were no significant main effects, despite a trend for the main effect of order ($F_{(1, 27.087)}=3.855$, p=0.060). Individuals' mean OT in the initial encounter predicted mean OT levels in the second (partner-stranger $r=0.97$, p<0.001; stranger-partner $r=0.57$, p=0.03).

### Hormonal cycles and OT levels

Of the 27 participants with full sample series included in the OT hormone analysis, 10 were not cycling (ovulation suppression by non-estrogen-based contraceptives, for example, copper intra-uterine devices), 13 were cycling naturally (using condoms as contraception, for example), and four did not report their cycle details. The mean OT of cycling participants fell within one standard deviation of noncycling participants for both partner and stranger (noncycling: 68.8 pg/ml±43.4 pg/ml for partner, 64.1±40.3 pg/ml for stranger; cycling: 62.9 pg/ml±31.1 pg/ml for partner, 48.6±21.3 pg/ml for stranger), indicating that cycle did not affect the variability of mean plasma OT. This was also the case for cycle phase among the cycling participants, of whom four were in the luteal phase (relatively stable but lower estrogen), two in the follicular phase (two early, two in the later days associated with sharply increased estrogen preceding ovulation), and seven were menstruating (decreasing to stable estrogen).

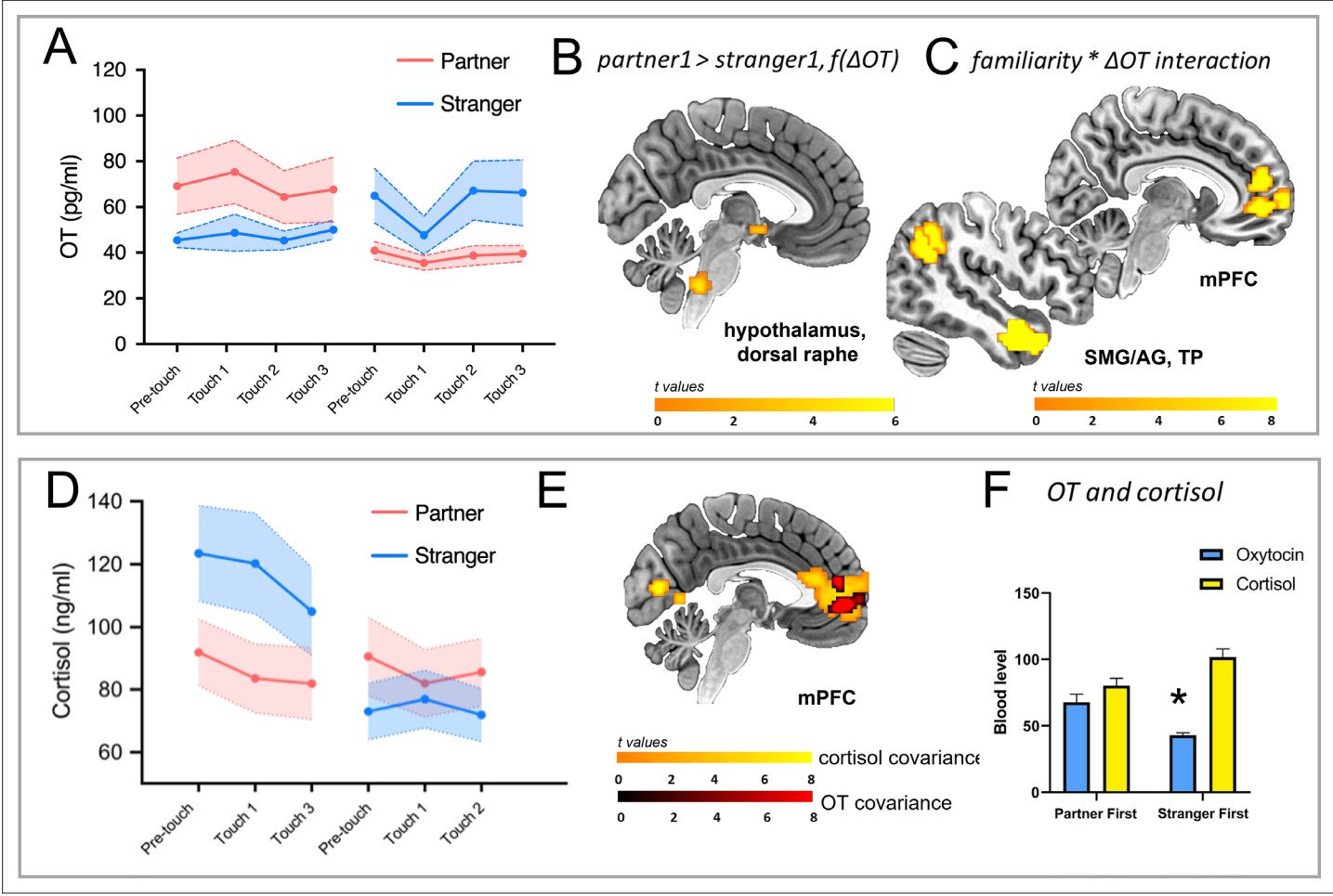

**Figure 2.** Endogenous hormone (OT and cortisol) changes and covariant brain responses. (**A**) Familiarity, order, and sample timepoint influenced plasma OT levels: familiarity, order, and sample timepoint interacted ($F_{(3, 183.180)}$=3.034, p=0.031) as did familiarity and order ($F_{(1, 183.169)}$=11.216, p=0.001), with increased OT when the partner was the interactant in the first encounter. The contribution of the sample timepoint lay in a dip and recovery during stranger touch only when preceded by partner touch (p=0.027). (**B**) OT-BOLD covariance in the hypothalamus and dorsal raphe was driven by a greater decrease for stranger touch during the initial encounter. (**C**) Familiarity and % OT change interacted in parietotemporal BOLD clusters along right SMG/AG, TP, and mPFC extending to ACC (3dLME model, p<0.002), reflecting more positive relationships between BOLD signal and OT change for partner (the higher the BOLD, the greater the degree of OT change). (**D**) Familiarity and order interacted in plasma cortisol levels ($F_{(1, 130)}$=54.89, p<0.001), with stranger touch eliciting a greater cortisol increase compared to partner touch, reflected in a main effect of familiarity ($F_{(1, 130)}$=15.67, p<0.001). There was also a main effect of sample timepoint ($F_{(2, 130)}$=3.16, p=0.045), with levels generally declining over the session. (**E**) BOLD signal change in regions including mPFC/ACC covaried as a function of cortisol levels, with partner >stranger (p<0.002), and a subset of mPFC voxels covarying with both OT (partner >stranger, second encounter) and cortisol (partner >stranger, initial encounter). (**F**) Mean values for OT and cortisol showed an interaction with familiarity and order factors over the session ($F_{(1, 178.355)}$=10.565, p=0.001), with higher OT but lower cortisol in the partner first condition as compared to stranger first. *OT = oxytocin, BOLD = blood-oxygen-level-dependent, f(ΔOT)=as a function of the change in OT, 3dLME = 3-dimensional linear mixed effects, SMG/AG = supramarginal gyrus/angular gyrus, TP = temporal pole, mPFC = medial prefrontal cortex, ACC = anterior cingulate cortex. All maps thresholded at p<0.002, corrected.*

## Cortisol levels

A total of six serial cortisol samples were collected during the course of the session. Cortisol showed a significant influence on partner or stranger (familiarity) but also their presentation order (interaction, $F_{(1, 130)}$=54.89, p<0.001; *Figure 2D*). Cortisol levels were higher when the stranger was presented first, compared to when he was presented second (p=0.008). In addition, stranger or partner (familiarity) as well as time point significantly influenced the participant's cortisol levels ($F_{(1, 130)}$=15.67, p<0.001 and $F_{(2, 130)}$=3.16, p=0.045, respectively). Cortisol levels were higher at the beginning of the touch session compared to the end (p=0.039) and stranger touch elicited a greater cortisol increase compared to partner touch (p<0.001).

## Combined effects of oxytocin and cortisol changes

Cortisol showed higher overall levels than OT (a main effect of hormone, $F_{(1, 180.593)}$=68.574, p<0.001; OT $M \pm SEM$: partner first: 67.781±6.019, stranger first: 42.959±1.816, p=0.017; cortisol $M \pm SEM$: partner first: 80.346±5.486, stranger first: 101.633±6.306, p=0.040). In addition, OT levels were higher but cortisol levels were lower in the partner first condition as compared to stranger first ($F_{(1,180.593)}$=28,751, p<0.001), and a statistical interaction indicated a mutual influence between familiarity and order ($F_{(1, 178.355)}$=10.565, p=0.001; also found in the two previous individual models; *Figure 2F*).

## Functional neuroimaging

### Linear mixed-effects modeling of factors order, familiarity, and site with OT covariate

In order to reveal blood-oxygen-level-dependent (BOLD) activations related to the order partner/stranger presentation (partner first, stranger first), the familiarity of the interactant (partner, stranger), the site of stimulation (arm, palm), and peak OT plasma levels (maximum percent change from baseline), we performed a linear mixed-effects modeling analysis (3dLME in AFNI). 3dLME was implemented because the analysis involved a between-subject factor (order), two within-subject factors (familiarity and site), and a quantitative variable or covariate (OT), modeled with random intercept and random slope (*Chen et al., 2013*). All analyses were thresholded at p=0.002 as per current AFNI recommendations (*Cox et al., 2017*).

Since OT response was influenced by the order factor, the peak OT values for each participant (n=26 complete datasets) were centered around the mean of each of the four conditions (partner first, stranger first, partner second, stranger second) prior to the analysis. This 3dLME analysis revealed cortical clusters in which familiarity, order, and OT showed mutual influences (a three-way interaction; *Table 1*). These clusters included the right superior occipital gyrus (SOG) extending into the right cuneus, and the left angular gyrus (AG). These interactions are explained by a more negative relationship between BOLD and OT change in the stranger first group, which reflects greater relative BOLD increases in individuals showing smaller percent change in OT levels. Specifically, whereas BOLD increase in SOG/cuneus showed a positive relationship in the partner first group (partner first, stranger second), these relationships were negative in the stranger first group, particularly for stranger. BOLD increase in AG showed positive or flat relationships with OT change in all conditions except for partner second in the stranger first group, which was negative (the higher the BOLD, the lower the OT change).

Familiarity and OT interacted in the bilateral angular gyrus (SMG/AG), inferior temporal gyrus (ITG), middle temporal gyrus (MTG), right temporal pole (TP), anterior cingulate cortex (ACC), and medial prefrontal cortex (mPFC, on the mid-orbital gyrus), among other regions (*Figure 2C*; *Table 1*). For all these areas, individuals showing greater OT increase during the functional run were also more likely to show higher BOLD during partner compared to stranger touch, reflecting a more positive relationship between BOLD signal and OT change for partner (the higher the BOLD, the greater the degree of OT change). A right hemisphere cluster encompassing inferior and middle temporal gyri (ITG and MTG) also showed a main effect of OT covariate (a positive relationship between BOLD and OT change; *Table 1*) as well as an interaction between order and OT (*Table 1*; see also Figure 4A), reflecting a more positive relationship between BOLD signal and degree of OT increase in the stranger first group.

Regardless of the degree of OT change, BOLD was overall greater for stranger than partner (a main effect of familiarity) in bilateral middle frontal gyrus (MFG) and right AG, among other regions (*Table 1*), reflecting generally higher means but a narrower range of variability for stranger than partner. BOLD was greater for the palm than the arm (a main effect of the stimulation site; *Table 1*) in the left postcentral gyrus (PoCG) and bilateral precentral gyrus (PrCG). No statistical interactions were observed between the site of stimulation and OT covariate.

### T-tests with OT covariate

To further compare BOLD activity across familiarity and order factors with OT as a covariate of interest, the following t-tests were performed: independent t-tests between partner first and stranger first and partner second and stranger second; and paired t-tests between partner first and stranger second and between stranger first and partner second (*Supplementary file 1*).

**Table 1.** Linear mixed-effects modeling with factors familiarity (partner, stranger), order (first or second encounter), with peak OT changes as a covariate.

All contrasts are thresholded at p<0.002, cluster-size thresholded at α = 0.05 FWE for n=27 complete functional datasets. For each cluster under each contrast heading, size in voxels, location, maximum *F* score, and MNI coordinates (x, y, z) are given.

**Main effect: Familiarity**
**StrangerPartner >**

| Cluster (size) | Peaks Locations | *F* (x, y, z) |
|---|---|---|
| #1 (362) | Left Middle Frontal Gyrus | 25.45 (-44, 52, 4) |
| | | 23.25 (-44, 25, 37) |
| | | 20.78 (-32, 52, 28) |
| | | 18.87 (-38, 61, 1) |
| | | 18.07 (-50, 43, 19) |
| | | 12.07 (-44, 13, 40) |
| | Left Inferior Frontal Gyrus | 24.79 (-56, 37, 10) |
| | | 20.27 (-59, 25, 22) |
| #2 (234) | Right Middle Frontal Gyrus | 33.33 (43, 46, 28) |
| | | 24.87 (34, 43, 40) |
| | | 18.90 (46 25, 37) |
| | | 16.93 (31, 28, 55) |
| | Right Superior Frontal Gyrus | 29.72 (22, 43, 37) |
| | | 16.03 (25, 58, 31) |
| | | 13.16 (19, 28, 40) |
| #3 (128) | Left Middle Occipital Gyrus | 28.69 (−17,−107, 4) |
| | | 18.25 (−26,−98, −5) |
| #4 (107) | Right Middle Frontal Gyrus | 21.16 (40, 46, 13) |
| | | 18.20 (40, 58, 13) |
| | | 14.56 (25, 61, 28) |
| | Right Superior Frontal Gyrus | 17.77 (31, 58, 19) |
| #5 (87) | Left Cerebellum | 21.42 (−41,−65, −38) |
| | | 19.14 (−29,−62, −35) |
| #6 (73) | Right Angular Gyrus | 19.18 (43, -65, 52) |
| | Right Middle Occipital Gyrus | 14.86 (34, -62, 37) |
| #7 (72) | Right Inferior Occipital Gyrus | 23.59 (25, -95, -8) |
| | | 15.83 (37, -95, -2) |
| #8 (65) | Right Cerebellum | 15.40 (34, -68, -32) |
| | | 13.24 (40, -74, -53) |
| | | 13.11 (40, -62, -50) |

**Main effect: Site**
**Palm >Arm**

| Cluster (size) | Peaks Locations | *F* (x, y, z) |
|---|---|---|
| #1 (1553) | Left Postcentral Gyrus | 204.14 (50, -29, 61) |
| | | 131.45 (−41,−26, 49) |

*Table 1 continued on next page*

*Table 1 continued*

**Main effect: Familiarity
StrangerPartner >**

| | | |
|---|---|---|
| | Left Precentral Gyrus | 126.07 (−35,−17, 64) |
| | Left Supplementary Motor Area | 58.04 (−8,−2, 52) |
| | Left Superior Parietal Lobule | 30.54 (−29,−59, 70) |
| | Left Superior Frontal Gyrus | 26.92 (−20,−2, 70) |
| | Right Supplementary Motor Area | 24.60 (10, -2, 52) |
| #2 (488) | Right Postcentral Gyrus | 122.40 (55, -23, 55) |
| | Right Superior Parietal Lobule | 17.13 (28, -53, 67) |
| #3 (444) | Right Cerebellum | 119.12 (19, -53, -23) |
| #4 (304) | Right Precentral Gyrus | 63.12 (37, -11, 67) |
| | Right Supplementary Motor Area | 28.37 (16, 7, 67) |
| | Right Superior Frontal Gyrus | 12.31 (19, -2, 55) |
| #5 (199) | Right Cerebellum | 76.11 (19, -59, -50) |
| #6 (89) | Left Cerebellum | 32.62 (−20,−53, 29) |
| #7 (72) | Right Middle Frontal Gyrus | 19.27 (31, 40, 22) |
| | Right Superior Frontal Gyrus | 17.35 (22, 46, 22) |

**Main effect: OT**

| Cluster (size) | Peaks Locations | $F$ (x, y, z) |
|---|---|---|
| #1 (93) | Right Middle Temporal Gyrus | 25.77 (49, 1, -20) |
| | Right Inferior Temporal Gyrus | 16.66 (61, -14, 35) |
| | | 14.87 (55, -17, -23) |

**Interaction: Familiarity*OT**

| Cluster (size) | Peaks Locations | $F$ (x, y, z) |
|---|---|---|
| #1 (585) | Right Superior Orbital Gyrus | 37.78 (19, 55, -5) |
| | Left Anterior Cingulate Cortex | 28.17 (-8, 49,−2) |
| | | 21.94 (-8, 49, 10) |
| | Right Anterior Cingulate Cortex | 24.01 (10, 49, 13) |
| | Left Superior Frontal Gyrus | 23.55 (-17, 61, 10) |
| | Left Mid Orbital Gyrus | 23.50 (1, 55, -2) |
| | Right Mid Orbital Gyrus | 19.24 (10,70, -11) |
| | Right Superior Medial Gyrus | 18.75 (13, 64, 16) |
| #2 (436) | Right Middle Temporal Gyrus | 36.99 (64, -17, -14) |
| | Right Medial Temporal Pole | 34.12 (40, 16, -32) |
| | Right Inferior Temporal Gyrus | 33.89 (49, -5, 29) |
| | | 21.52 (61, -14, 35) |
| | | 19.36 (52, -17, -26) |
| #3 (261) | Right Angular Gyrus | 26.54 (46, -59, 34) |
| | | 25.40 (58, -56, 25) |
| | | 22.59 (43, -53, 25) |

*Table 1 continued on next page*

*Table 1 continued*

**Main effect: Familiarity
StrangerPartner >**

| Cluster (size) | Peaks Locations | F (x, y, z) |
|---|---|---|
| | | 21.22 (58, -62, 37) |
| | Right Middle Occipital Gyrus | 15.73 (43, -74, 31) |
| #4 (185) | Left Inferior Temporal Gyrus | 32.65 (-41, 4,–38) |
| | | 23.30 (−56,−5, −38) |
| | Left Middle Temporal gyrus | 24.21 (-47, 4,–26) |
| | Left Medial Temporal Pole | 13.62 (-53, 16,–32) |
| #5 (144) | Left Angular Gyrus | 35.84 (−50,−65, 49) |
| | | 21.85 (−56,−56, 34) |
| | Left Inferior Parietal Lobule | 13.96 (−41,−59, 58) |
| #6 (138) | Left Cerebellum | 25.48 (−53,−59, −35) |
| | | 15.48 (−35,−77, −35) |
| | | 15.48 (−35,−77, −35) |
| | | 14.08 (−50,−68, −44) |
| | | 12.64 (−47,−74, −32) |
| | | 12.27 (−47,−62, −48) |
| #7 (92) | Right Superior Frontal Gyrus | 39.89 (22, 28, 61) |
| | | 27.84 (22, 16, 67) |
| #8 (79) | Left Middle Frontal Gyrus | 25.33 (-41, 25, 52) |
| | | 23.11 (-35, 40, 43) |
| #9 (62) | Right Inferior Temporal Gyrus | 24.67 (43, -11, -32) |
| | Right ParaHippocampal Gyrus | 15.74 (28, -8, 35) |
| | Right Fusiform Gyrus | 14.45 (31, -2, -44) |

**Interaction: Order*OT**

| Cluster (size) | Peaks Locations | *F* (x, y, z) |
|---|---|---|
| #1 (195) | Right Inferior Temporal Gyrus | 26.44 (43, -11, -32) |
| | | 15.30 (61, -14, -35) |
| | Right Middle Temporal Gyrus | 22.79 (49, 1, -20) |

**Interaction: Familiarity*Order*OT**

| Cluster (size) | Peaks Locations | *F* (x, y, z) |
|---|---|---|
| #1 (111) | Right Superior Occipital Gyrus | 29.02 (25, -98, 19) |
| | Right Cuneus | 22.76 (19, -104, 10) |
| #2 (81) | Left Angular Gyrus | 25.20 (−47,−59, 34) |

## Partner first vs. stranger first

Raphe nuclei (whole brain) and hypothalamus (ROI analysis) showed a greater positive covariation between BOLD and OT change for partner touch compared to stranger touch in the first run (*Figure 2B*; *Supplementary file 1*). To explore whether BOLD signal changes in the hypothalamus covaried with OT levels with respect to familiarity, we first explored whether all subjects included in the whole-brain analysis had representative data in those voxels to be included in a region of interest (ROI) approach. To do this, we used the Neurosynth database's 'association test map' for the search

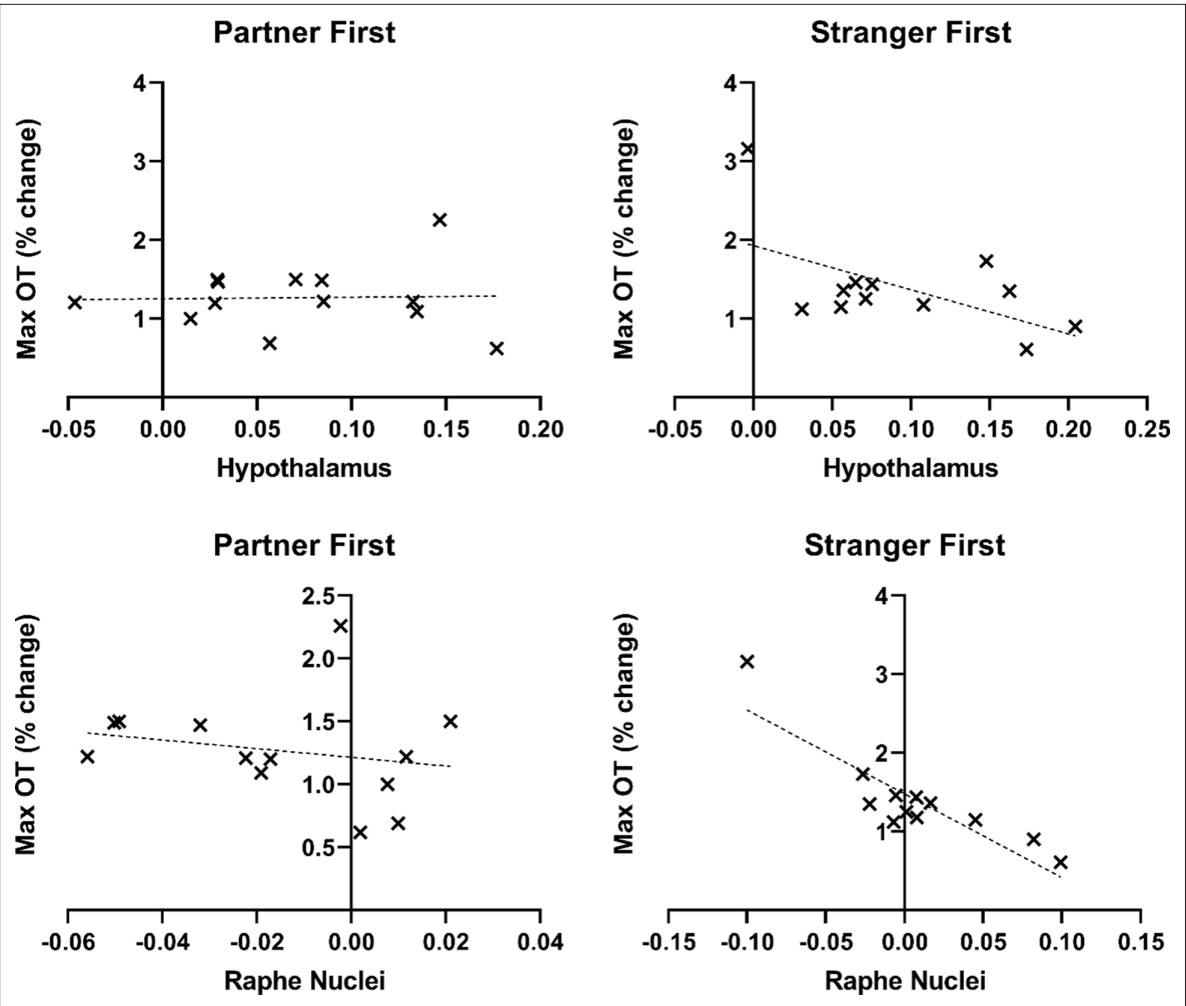

**Figure 3.** Scatterplots showing the relationship between blood-oxygen-level-dependent (BOLD) signal and change in oxytocin (OT) during the first encounter in the hypothalamus and Raphe nuclei. The linear mixed-effects modeling weighted the BOLD signal with an OT covariate (not visualized in the scatterplot); These showed that all standardized residuals fell within ±3 standard deviations from the trendline.

term 'hypothalamus' (https://neurosynth.org/analyses/terms/hypothalamus). This 1565-voxel map also encompassed regions outside the hypothalamus (e.g. thalamus and brainstem), so a restricted threshold was applied to include only voxels located in the hypothalamus. The size of the final ROI was 111 voxels, and this was used in a subsequent SVC analysis including 25 of the initial 27 participants. Two participants (one per group) were excluded as they had data for fewer than 50% of the ROI voxels in at least one of the two functional runs (partner touch, stranger touch). Within this cluster, a subset of 15 voxels showing greater increases for partner touch than stranger touch survived stricter cluster correction at p=0.002. OT-BOLD covariance in the hypothalamus and dorsal Raphe showed a greater negative correlation for stranger touch during the initial encounter (*Figure 3*).

The linear mixed-effects modeling weighted the BOLD signal with an OT covariate. However, it is not possible to represent this weighting when plotting the beta values extracted from the resulting cluster (see scatterplots in *Figure 3*). Thus, outliers on either measure (OT or beta value) will appear as outliers in the scatterplot, despite not necessarily being outliers with respect to the population trend of the relationship between OT and BOLD. To determine whether a given data point was an outlier in this regard for the hypothalamus region-of-interest, we, therefore, examined the standard deviations of the residuals in the model (distance above and below the trendline), by entering the values into a regression model with beta values as an independent variable and OT (maximum % change from baseline) as a dependent variable. The standardized residuals were calculated in standard deviation units for each group (partner first and stranger first). These showed that all standardized residuals fell

within ±3. The regressions were significant at p<0.05, for both hypothalamus (p=0.003 for partner first, p=0.03 for stranger first) and Raphe nuclei.

### Partner second vs. stranger second

In parietotemporal clusters, the higher an individual's BOLD, the higher her OT levels when receiving partner touch in the second run, as compared to those receiving stranger touch in the second run (in whom this OT-BOLD covariation was less positive). These parietotemporal clusters were also seen in the interaction maps in which OT interacted with familiarity (right AG, right TP; *Figure 2C*) and order (right MTG). Additional clusters were revealed in the right superior temporal gyrus (STG), among other activations (*Supplementary file 1*).

### Partner first vs. stranger second

There was no activation difference between partner and stranger touch in the partner first group.

### Stranger first vs. partner second

There was no activation difference between partner and stranger touch in the stranger first group.

## Linear mixed model of factors order and familiarity with cortisol covariate

As in the OT analysis, we searched for brain correlates of touch interactions as influenced by familiarity (partner, stranger), order (partner first, stranger first), and cortisol plasma level, performing a linear mixed-effects modeling analysis (3dLME in AFNI; n=26). Again, we adopted the model with random intercept and random slope, and cortisol mean values (percent change from baseline) for each participant were centered around the mean of each of the four conditions (partner first, stranger first, partner second, stranger second) prior to the analysis. Results did not show any significant main effect or interaction.

## T-tests with cortisol covariate

As for OT, in order to compare neural activity across the familiarity and order factors with cortisol as a covariate of interest, we ran the following t-tests: paired t-tests between 'partner first' and 'stranger second' and between 'stranger first' and 'partner second'; independent t-tests between 'partner first' and 'stranger first' and 'partner second' and 'stranger second' (*Supplementary file 2*).

### Partner first vs. stranger first

In the first run, the two groups did not differ in terms of activation according to the identity of the stroker.

### Partner second vs. stranger second

In the second run, there was no differential activation between partner first and stranger first groups related to the identity of the stroker.

### Partner first vs. stranger second

For participants who had partner touch first, partner compared to stranger touch covaried with cortisol in the left ACC, right SMG, bilateral ventromedial prefrontal cortex, bilateral calcarine gyrus, right lingual gyrus, left TP, and left PO (all at p=0.002). These regions showed a more negative relationship between BOLD and cortisol for partner (individuals with higher BOLD had higher cortisol).

### Stranger first vs. partner second

When comparing brain activity in the two runs in the group of participants who had stranger touch first, we found no cortisol-related differences in brain activity between partner and stranger touch.

## OT regressor: Exploratory analysis

To discover activation corresponding with the overall temporal pattern of the endogenous OT response over the sample series in each functional run, we used each participant's serial plasma OT levels to create a custom regressor for each individual. We assumed that any central-to-peripheral effects of OT release would be detectable retrospectively by modeling the plasma OT sample points 'backward,'

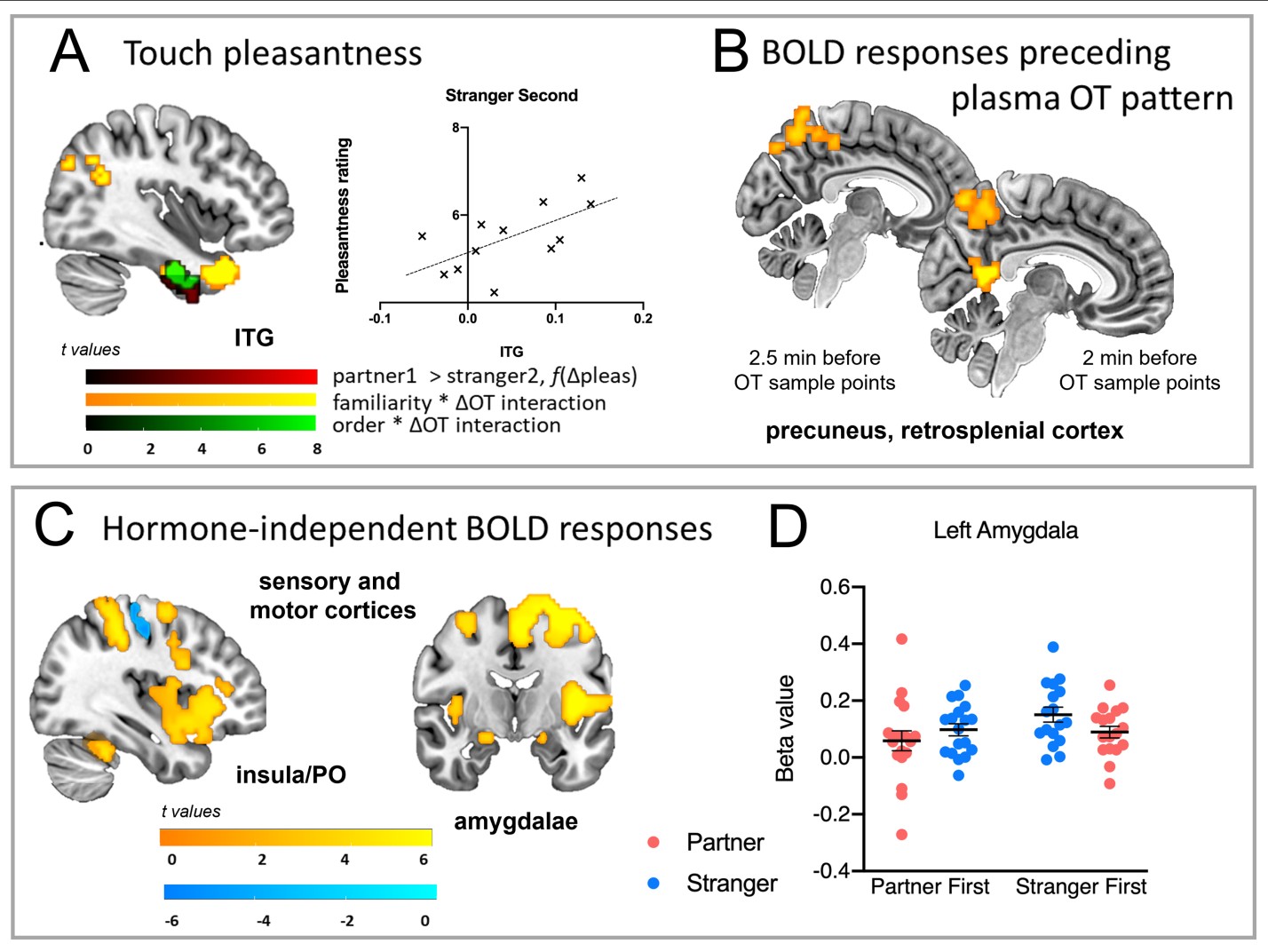

**Figure 4.** Hormone-independent BOLD responses, BOLD covariance with temporal OT pattern, and BOLD covariance with pleasantness ratings.
(**A**) The temporal pattern of OT-BOLD changes in bilateral precuneus preceded sampling by 2.5 min, with retrosplenial cortex also emerging at 2 min, showing greater OT-BOLD covariance for stranger touch during the initial encounter. (**B**) ITG/TP was sensitive to differences in touch pleasantness ratings (red) for partner and stranger (mean pleasantness partner >stranger, p<0.001), with BOLD increasing with partner vs stranger pleasantness differences (Δpleas) during stranger touch in the second encounter (scatterplot). ITG/TP clusters also showed an interaction between familiarity and ΔOT (yellow), and between partner/stranger presentation order and ΔOT (green) BOLD here was greater in individuals with smaller OT change during the stranger-second condition. (**C**) BOLD changes in somatosensory and insular and PO cortices, as well as bilateral amygdalae, across all touch conditions, independently of familiarity of the person delivering touch, order, and OT levels (all *p*s <0.002). (**D**) Beta values reflecting the BOLD signal change in the left amygdala sensitive to partner-stranger differences (main effect of familiarity, $F_{(1,16)}$ = 5.8, *P*=0.02), greater for the stranger in the first encounter. *BOLD = blood-oxygen-level-dependent, PO = posterior operculum, OT = oxytocin, ITG = inferior temporal gyrus, TP = temporal pole, mPFC = medial prefrontal cortex, f(Δpleas)=as a function of the change in pleasantness ratings. All maps thresholded at p<0.002, corrected.*

in order to search for any BOLD activity which both preceded and tracked the observed pattern of OT changes. Points between the multiple samples were linearly interpolated and the resulting function was convolved with the canonical hemodynamic response function (HRF). N=23 participants had complete data series for both functional runs. For this 'backward-looking' regressor, we explored time lags of 1, 1.5, 2, 2.5, and 3 min to capture potential touch-evoked central modulation corresponding to the peripheral changes in plasma OT observed after these various delays, assuming that central activity preceded peripheral OT changes.

This analysis revealed brain areas showing significant interactions between familiarity and order at both 2 and 2.5 min, with higher BOLD during stranger first than partner first, but no difference between stranger second and partner second (***Figure 4B, Supplementary file 3***). No activations

were observed for the remaining time lags. *2 min.* Bilateral precuneus, right SMG, left inferior parietal lobule, right posterior cingulate cortex, right postcentral gyrus, and right parietal operculum showed increased BOLD corresponding to the pattern of changes in OT levels observed 2 min after the activation (p=0.002; *Figure 4B*, *Supplementary file 3*). *2.5 min.* Two and half minutes before observed changes in OT levels, increased BOLD in the bilateral precuneus and right paracentral lobule corresponded with the temporal pattern of OT changes (all at p=0.002, *Figure 4B*, *Supplementary file 3*).

## Hormone-independent analysis

### T-test with touch pleasantness covariate

To discover regions in which BOLD activation covaried with changes in touch pleasantness ratings, we performed t-tests between partner and stranger for each presentation order (partner first or stranger first) with the difference in pleasantness ratings (partner minus stranger) as a covariate of interest. There was no resulting activation in the stranger first group. In the partner first group, a cluster in ITG was revealed, which overlapped with the ITG/TP clusters in which OT interacted with familiarity and order, respectively (*Figure 4A*, *Supplementary file 4*). Here, individuals that showed higher BOLD activation for partner compared to stranger also showed the largest difference in ratings between partner and stranger touch. When the stranger delivered touch in the second encounter, ITG activation increased with touch pleasantness during the run (*r*=0.65, p=0.003; *Figure 4A*).

### Conjunctions

To identify common activation across all touch conditions regardless of familiarity, order, or hormone levels, we created single condition maps by contrasting each condition against baseline activity at p<0.002 (*Figure 4C*, *Supplementary file 5*).

### Partner first ∩ stranger first ∩ partner second ∩ stranger second

Brain areas showing increased activity irrespective of condition were: bilateral supramarginal gyrus (SMG), bilateral postcentral gyrus, left precentral gyrus, left superior parietal lobule (SPL), left parietal operculum (PO), bilateral supplementary motor area (SMA), right inferior frontal gyrus (IFG), bilateral anterior insula, right cerebellum, right supplementary motor area (SMA), and right amygdala. There was a common deactivation in the right precentral gyrus. While BOLD signal change in the left amygdala was high across conditions, it was also sensitive to partner-stranger differences (main effect of familiarity, $F_{(1,16)}$ = 5.8, p=0.02), greater for the stranger in the first encounter.

## Discussion

A single social interaction with a familiar partner is just one instance of many over the course of the relationship. On the other hand, today's familiar friend was yesterday's stranger: an interaction with a person one has never met can lay the groundwork for future interactions. Neural and hormonal changes elicited during successive social interactions must, therefore, not only be able to maintain stability with respect to established social relationships (*Quintana and Guastella, 2020*) —such as with a romantic partner—but must also be adaptable in the face of new or less certain relationships, such as meeting a new individual. The present findings shed light on the participation of OT in brain-OT covariation during social encounters with both familiar and unfamiliar individuals. They imply that OT and the brain can flexibly coordinate and calibrate responses depending on whom an individual is currently socially interacting with, and with whom the individual has recently interacted.

The most general finding was that touch-mediated social interactions in human females elicited endogenous OT and brain responses in a covariant manner. Beyond this, OT and neural changes were modulated by the familiarity of the person delivering touch, as well as the recent history of social interaction. The effect of these contextual factors on within-subject endogenous OT changes manifested in a mutual influence between the familiarity of the social interactant (partner or stranger) and the order of his presentation over two immediately successive social interactions (partner then stranger, or stranger then partner). This influence was driven by a greater increase in plasma OT responses for the stranger following partner touch, in the absence of a corresponding increase for partner touch following stranger touch (*Figure 2A*). This dependence of OT responses on both familiarity and presentation order is consistent with evidence for cumulative effects of central OT exposure

over repeated interactions with specific individuals (*Burkett et al., 2016*), as well as the established context-sensitivity of OT effects in nonhuman mammals (*Bartz et al., 2011*).

Social familiarity and presentation order also influenced the timecourse of the eight plasma samples collected during the experimental session, with this influence driven by OT responses to stranger touch following partner touch. The OT increase for stranger touch in this condition did not show a stable rise but rather dipped to below-baseline levels across participants before recovering to above-baseline levels by the end of the social encounter. A tentative interpretation of this pattern is that the initial partner encounter may have introduced a bias for OT increase during the subsequent stranger encounter, though not as a sustained carryover from the preceding partner interaction. In contrast, plasma OT remained at baseline levels when the experimental session began with an encounter with an unfamiliar stranger. The recovery of OT in the stranger-second condition could thus reflect a facilitation of underlying endogenous release mechanisms following the prior partner interaction. Overall, these endogenous OT changes may reflect mechanisms that selectively bias the way social stimuli are processed in the central nervous system during social interactions, with high dependence on contextual information.

## BOLD correlates of endogenous OT changes

The mutual influences observed in OT levels among social familiarity and partner/stranger presentation order were mirrored in OT-covariant BOLD responses in key parietotemporal and frontal regions, on a whole-brain level (*Figure 2C* and *Figure 2E*). In particular, OT changes showed mutual influences with both familiarity and presentation order in the temporal pole (TP). In SMG/AG, IFG, mPFC/ACC, and superior frontal gyrus, OT-brain covariation was affected by the familiarity of the individual delivering the touch. These parietotemporal and medial prefrontal regions have been implicated in individual intranasal-OT studies of partner-stranger interactions (*Kreuder et al., 2017*; *Kreuder et al., 2019*; *Scheele et al., 2012*; *Scheele et al., 2013*), animate visual social stimuli (*Lancaster et al., 2015*), as well as in meta-analyses of intranasal-OT fMRI activation (*Wang et al., 2017*; *Zink and Meyer-Lindenberg, 2012*) and even in tactile foot massage (*Li et al., 2019*).

Co-modulation between OT changes and BOLD in parietotemporal pathways may reflect updating of contextual information, possibly enhancing the salience of incoming sensory signals (*Johnson et al., 2017*; *Shamay-Tsoory and Abu-Akel, 2016*; *Sripada et al., 2013*) or of personally-relevant stimuli *Alaerts et al., 2021* following the initial encounter.

SMG/AG, MTG, and ITG/TP all showed a more positive relationship between BOLD and the degree of OT increase during partner compared to stranger encounters. BOLD in ITG/TP was also sensitive to differences in touch pleasantness for partner and stranger, with higher BOLD for stranger second the greater the partner-stranger rating difference (*Figure 4A*).

In a right AG cluster overlapping the SMG/AG cluster, the generally positive relationship between BOLD and OT change selectively modulated in a negative direction when the partner administered touch in the second encounter. Here, BOLD was *greater* in individuals with *smaller* OT changes during the partner-second condition. Conversely, BOLD in an ITG/TP cluster was greater in individuals in the partner-first group in whom OT change was smaller (*Figure 4A*, green), likewise indicative of selective modulation. This implies that initial OT-brain biases by partner or stranger may give rise to recalibration processes, acting either against (i.e. partner second for AG, stranger second for ITG/TP) or with (all other conditions) the degree of endogenous OT changes. A speculative interpretation is that OT-brain co-modulation may find and maintain a stable response profile following partner touch, even after an unfamiliar stranger is presented, whereas such recalibration can be more sluggish following an initial encounter with the stranger. This may reflect an enlistment of parietotemporal regions in gain control mechanisms (*Grinevich and Ludwig, 2021*)—more akin to a dimmer switch than an on-off button— with the 'dimmer' tuning from a wider, more flexible response range under higher contextual certainty (partner first) to a narrower, less flexible range under lower contextual certainty (stranger first).

In the initial touch encounter, the preferential increase in plasma OT levels for the partner is consistent with the familiarity-dependent effects of OT in rodents (*Burkett et al., 2016*). This basic partner-stranger difference was reflected in hypothalamus OT-covariance as a function of individual changes in plasma OT levels (*Figure 2B*). This is in accord with the conserved mammalian neuroanatomy of central OT release, in which hypothalamic nuclei, particularly the PVN, synthesize OT and stimulate

its release in the brain (*Burbach et al., 2005*; *Knobloch et al., 2012*; *Mitre et al., 2018*; *Oettl et al., 2016*; *Wang et al., 2022*). In rats, stroking touch increases Fos protein expression in PVN (*Okabe et al., 2015*), and recent evidence from freely-interacting female rats indicates that a population of parvocellular OT neurons in PVN is selectively tuned to social touch stimulation (*Tang et al., 2020*) and also that this is associated with subsequent changes in levels of plasma OT (*Tang et al., 2020*). In the present study, BOLD changes in dorsal Raphe nuclei were also covaried with plasma OT (*Figure 2B*). Specifically, when the stranger touched first, BOLD in both the hypothalamus and dorsal Raphe was greater the lower the mean OT across individuals (*Figures 1 and 3*), whereas no such modulation of OT-BOLD covariance was observed for partner. This may reflect descending modulatory influence on afferent touch signals from the periphery, manifesting here as a negative relationship between BOLD and OT during stranger touch (the BOLD signal cannot distinguish hemodynamic activity resulting from excitation or inhibition).

Central effects of IN-OT have consistently been found ~45 min post-administration (*Martins et al., 2020*; *Sripada et al., 2013*; *Valstad et al., 2017*), but there is less direct evidence about the time-course of central endogenous OT release into the periphery in humans. We, therefore, developed an exploratory regressor based on the serial pattern of individuals' OT levels. Assuming a mechanism in which plasma OT changes were affected by central release in the brain and thus came after it in time, this regressor allowed us to look 'backward' from the temporal pattern of the plasma OT sample series to any preceding hemodynamic activation that tracked with this pattern. The pattern-covariant engagement of the precuneus at 2.5 min preceding sampling, and of precuneus, retrosplenial cortex, and mPFC at 2 min, is within the frame of the half-life of OT in the blood (*Pow and Morris, 1989*) and may reflect events surrounding central OT release (*Qin et al., 2009*; *Figure 4B*). Although the present results lack the sufficient temporal and causal resolution to address this, it is possible that any descending OT influence from the brain to the periphery instates a 'reafferent loop' in which central-to-peripheral changes can, in turn, influence incoming sensory information, perhaps at the level of spinal and/or brainstem mechanisms.

Retrosplenial cortex projects to mPFC (*Margulies et al., 2009*), while precuneus is functionally connected to the AG region showing interactions between OT and familiarity (*Figure 2C*; bilateral AG extending to left SMG) and OT and order (right AG), described above. IN-OT administration can also induce changes in functional connectivity between precuneus and AG (*Kumar et al., 2020*). These findings imply that the precuneus, retrosplenial cortex, and mPFC may act as arbiters of activation in parietotemporal and limbic networks, potentially influencing responses to social touch via contextual integration and affecting regulation processes. For example, the covariant relationship of ITG activation with plasma OT changes as well as touch pleasantness ratings (*Figure 4A*), suggests a potential role for this region in maintaining receptivity to touch following contextual shifts. Such OT-dependent neural dynamics may play a critical role in calibrating social receptivity, especially over multiple social encounters. It is not possible to determine the directness or direction of any corticocortical influences from these exploratory findings, however.

Here, mean plasma cortisol levels were higher for stranger than for partner during the first encounter (*Figure 2D*) and these decreased as mean OT levels increased during the course of the experiment. OT has been implicated in stress regulation via corticotropin-releasing-hormone (CRH) pathways that result in cortisol changes, and so may act as a physiological regulator of acute stress-related responses (*Ditzen et al., 2009*; *Grewen et al., 2005*; *Petersson and Uvnäs-Moberg, 2003*; *Vargas-Martínez et al., 2014*; *Winter and Jurek, 2019*). Here, plasma cortisol levels covaried with BOLD for partner in the initial encounter in several regions, including a mPFC activation that contained an OT-sensitive cluster (*Figure 2E*). Further, activation in both mPFC, implicated in cortical-amygdala signaling following IN-OT, and superior temporal gyrus (STG), implicated across a range of multisensory integration, including touch (*Davidovic et al., 2016*; *Kaiser et al., 2016*; *Voos et al., 2013*), decreased for initial stranger touch as a function of cortisol, compared to partner.

Taken together, these selective hormone-brain changes support the view that endogenous OT's role in human social interaction is heavily modulated by contextual factors (*Bartz et al., 2011*), and provides further evidence that this role can involve modulation in a positive or a negative direction, depending on the situation (*Hovey et al., 2016*; *LoParo et al., 2016*; *Rickenbacher et al., 2017*). For example, in prairie voles, higher levels of endogenous OT can mediate prosocial grooming of stressed others (*Burkett et al., 2016*), but optogenetic manipulation of the same PVN OT neurons in

freely-behaving mice can result in either prosocial or antagonistic behavior (*Yu et al., 2022*). OT and OT receptor genotypes have also been shown to play a role in antagonistic social behaviors such as the defense of offspring in rats (*Rickenbacher et al., 2017*) and aggression in rodents and humans (*Hovey et al., 2016*; *LoParo et al., 2016*).

## Hormone-independent BOLD changes

BOLD responses in several key regions were independent of plasma OT or cortisol changes (*Figure 4A*), suggesting an absence of direct, covariant modulation with respect to these endogenous hormones. The bilateral amygdalae were activated in a general fashion across all encounters (*Figure 4C*; *Supplementary file 5*), with the left amygdala selective for stranger touch, particularly in the first encounter (*Figure 4D*). Amygdala activation has been widely implicated in human IN-OT studies (*Kirsch et al., 2005*; *Liu et al., 2019*; *Motoki et al., 2016*; *Sripada et al., 2013*), yet with inconsistent reports of the direction of BOLD changes, implying a dependence on experimental and methodological factors. In mice, OT receptor-expressing neurons in the medial amygdala have been found to mediate olfactory-based social familiarity effects (*Ferguson et al., 2001*). Here, though, stranger touch activated the amygdala in a general fashion (*Figure 4C*; *Supplementary file 5*), supporting the proposal that its prominent role in human and nonhuman primate OT studies may be indirect (*Eckstein et al., 2017*; *Gothard and Fuglevand, 2022*; *Putnam et al., 2018*).

IN-OT has been observed to increase the pleasantness of touch (*Chen et al., 2020*), and individuals with higher salivary OT levels have reported greater touch pleasantness (*Portnova et al., 2020*). Likewise, CT afferent nerve responses in the skin have been associated with subjective touch pleasantness of caress-like stimuli on a group level (*Löken et al., 2009*), though pleasantness ratings for touch show individual variability and lack of specificity (*Croy et al., 2021*; *Sailer et al., 2020*). However, this experiment provided no supporting evidence for a putative relationship between OT and CT afferent nerve activity associated with affective touch. Primary somatosensory cortex showed selective activation for touch on the palm. Yet no OT-BOLD covariance was observed here or in arm-specific regions such as the posterior insula/PO, which might have indicated a link between stimulation of CT-rich skin and endogenous OT. In rodent models, oxytocin receptor (OXTR) expression has so far not been identified in dorsal horn neurons of the spinothalamic tract projecting to the specific thalamic pathways putatively shared by CT afferents (*Gauriau and Bernard, 2004*; *Moreno-López et al., 2013*; *Nersesyan et al., 2017*). On a subjective level, participants found palm touch from a stranger less pleasant during the initial encounter, which was not predicted. This potentially reflects a functional difference in the palm's prominent role in active sensorimotor exploration (*Morrison, 2022*), and perhaps potentiation of approach or withdrawal from others' touch. Further research is, therefore, needed to explore any functional link between CT afferents and OT.

## Limitations and future directions

Measurement of peripheral OT in humans comes with caveats, as does its relationship with central mechanisms of release. Different methods for detecting plasma OT have yielded different and sometimes uncorrelated sets of value ranges, with measurement issues centering around the selectivity with which testing components bound or unbound protein, or whole or fragmentary molecules. Cerebrospinal fluid (CSF) has consistently been found to correlate more strongly with brain OT levels than does plasma OT, while CSF and plasma OT measurements have shown weak or no correlation (*Caicedo Mera et al., 2021*). However, at least a proportion of inconsistencies in reported findings may depend on a historical tendency to investigate basal levels, rather than acute stimuli more likely to evoke coordinated, biologically-meaningful responses across the central and peripheral nervous systems. Such responses may also include the bioactivity of OT fragments (*Uvnäs Moberg et al., 2019*). In this study, though, within-subject serial sampling allowed assessment of evoked OT changes with respect to individual baselines. The covariation of these changes and their temporal patterns with BOLD points to a relationship between the central and peripheral effects of ecologically-evoked endogenous OT, but cannot directly demonstrate it.

Most human studies manipulating OT (usually via nasal administration) have been performed in male populations. In contrast, the present study used a 'female-first' strategy which moves to redress this imbalance *Shansky and Murphy, 2021*; likewise, recent research in rodents has focused on OT-touch mechanisms in female samples (*Tang et al., 2020*; *Yu et al., 2022*). Here, we found no

effect of the cycle phase on evoked OT changes. An important question for future research is whether, and to what extent, these results in human females generalize to males, especially with regard to any familiarity-dependent bias in endogenous OT. Another potential sex difference may lie in the relationships between OT, cortisol, and their covariation in mPFC.

In everyday life, we humans must navigate a complex and ever-changing social terrain, with some stable elements (for example, established relationships) and other less-stable ones (new or uncertain relationships). This presents a challenge for maintaining the stability of existing social bonds on the one hand, yet also establishing and calibrating newer social relationships on the other hand. These findings suggest a role for OT-brain covariation in such adaptive responses. A positive social interaction context (such as a pleasant touch interaction with one's partner) may selectively bias the system towards a certain shorter-term neurohormonal response profile, whereas a less certain or less positive social context (such as an unusual interaction with a stranger) may bias it towards a different profile. For example, this could mean that starting the day with a positive social interaction can set up a virtuous circle that perpetuates itself through the day's social interactions; whereas an uncertain or negative interaction could bias one's responses towards remaining dampened. Such differential outcomes may potentially influence neural processing and behavior in longer-term social interactions. An important avenue for future research will be to investigate the behavioral effects of these neural and physiological differences, especially with respect to social relationships over time.

## Conclusions

These findings offer a methodological and conceptual bridge between stimulus-driven and context-sensitive frameworks of endogenous OT modulation of the brain during social interactions. Touch-mediated social interactions evoked changes in endogenous OT, indicating the importance of the stimulus. Yet these responses were nevertheless influenced by specific features of social context, with the plasma of OT levels showing biases depending on the familiarity of the interacting person and the recent history of interaction. Such adaptive responses could reflect a gain-control-like role for OT-brain neuromodulation, comparable to a dimmer switch, which can effectively preserve stability with respect to established social relationships while also allowing for a change in new ones (possibly via increases or decreases in inhibitory influence). Across successive social encounters, such modulatory mechanisms may calibrate neural and behavioral receptivity, whether mediated by touch or another channel such as vision or speech. Network hubs in parietotemporal pathways, alongside precuneus and retrosplenial cortex, may be key for turning the 'dimmer' of OT-brain processing up or down depending on past and current social context.

## Materials and methods
### Participants

42 females in committed heterosexual romantic relationships of at least one year (age *M*=24.6 years, SD = 4.6, *Supplementary file 6*), participated in the study with their male partners (age *M*=26.8 years, SD = 6.0). Female participants were included if they were between 19–40 years, were not pregnant or breastfeeding, did not use estrogen-based contraceptives, and were not undergoing current or recent hormone therapy. The female in the couple participated in fMRI scanning and provided plasma samples, while the male partner administered touch during the experiment.

This study took a female-first strategy (*Shansky and Murphy, 2021*), in order to limit any confounding sex-specific effects and between-sex variability in OT and cortisol responses, and also in light of the predominance of males in human OT studies. We collected cycle phase self-report (with the aid of apps for most cycling participants) to rule out any variability which might be associated with relative estrogen increases during the late follicular phase, in light of the evidence for OT-estrogen interactions in the context of sexual responses and parturition (e.g. *Becker et al., 2016*; *Filippi et al., 2003*; *Salonia et al., 2005*, but see *Itoh and Arnold, 2015*; *Prendergast et al., 2014* on the lack of evidence for effects of cycle-related variability on neural and physiological measures and gene expression in mice and humans).

Ethical approval was obtained from the Regional Ethical Review Board in Linköping, Sweden. All participants gave informed consent in accordance with the Declaration of Helsinki and were compensated at 400 SEK (~45 USD)/h.

## Procedure

An indwelling magnet-safe catheter was inserted into the cubital vein of the female participants' left arm 40–60 min before the scanning session, to reduce the possibility of short-term effects of the challenge of needle insertion on plasma hormone levels during the main experiment. This took place in the same building as the MR suite.

In the ensuing 45–60 min the participant and her partner filled out two questionnaires (separately); the Couples Satisfaction Index (CSI; *Funk and Rogge, 2007*) and the State-Trait Anxiety Inventory (STAI; *Spielberger, 1970*). The CSI is a 32-item scale designed to measure satisfaction in a relationship. The total score is the sum of responses across all 32 items and can range from 0 to 161. Scores below 104.5 suggest relationship dissatisfaction, while higher scores suggest greater levels of relationship satisfaction (*Funk and Rogge, 2007*). The STAI was used to measure the participants' self-assessed anxiety. There are two subscales in STAI; one that determines state anxiety (STAI-S) and one that measures trait anxiety (STAI-T). The questionnaire contains 40 items in total and each item is scored on a 4-point Likert scale, with a total score range of 20–80 (*Spielberger, 1970*). The median alpha reliability coefficients in healthy individuals for the STAI questionnaire (STAI-S and STAI-T) are 0.92 and 0.90, respectively (*Spielberger, 1970*). Missing data in both the CSI and STAI questionnaires (maximum two missing values) was handled with hot deck imputation (*Myers, 2011*).

After completing the questionnaires, participants and their partners received instructions about the experimental procedure together. They were requested not to touch each other during this period. They were also informed of the presentation order (partner or stranger first) and were briefly introduced to the stranger (a male employee working in the lab), before going into the scanner. Before the partner/stranger entered the MR room for a functional run, the participant was informed over the audio system about who would be entering the room. This was to reduce any uncertainty as to the identity of the person delivering the touch, to avoid the risk of increasing psychological distress for the participant, as well as any potential increase in variance in the measures collected. Partners that were to deliver touch in the second run waited in a furnished staff break room until they were accompanied to the MR room.

## Experimental design and session structure

The participants performed both sessions during their single visit to the lab, i.e., they received a touch from both partner and stranger during the same visit. The experimental paradigm implemented a 2 × 2 × 2 factorial design: 'familiarity' (partner/stranger, blocked by run), 'order' (partner or stranger in the first encounter, counterbalanced), and 'site' of the touch (arm or palm). Each session included one 7 min functional run that consisted of twelve 12 s touch stimulation trials, with arm and palm stimulation pseudorandomized within the run, and a jittered 21–30 s intertrial interval. The first functional run was preceded by a T1-weighted anatomical scan. There was a~27 mn interval between the first and second session, during which the participants remained in the scanner, and T2-weighted anatomical scans, resting state, and diffusion data (not analyzed here) were collected (*Figure 1B*).

Touch stimulation was delivered manually by the male interactant (partner and stranger). Caressing strokes were delivered to the right dorsal arm or palm, with timing and touch site guided by auditory cues via headphones (*Figure 1A*). The interactant was positioned beside the scanner bore on the right side of the participant. In the initial encounter (the first of the two functional runs), participants either received a touch from their partner or the stranger, and vice-versa in the second encounter (n=24 and n=18, respectively). In the last 7 s of each trial, the participant rated the pleasantness of the touch on a visual analog scale (VAS), anchored with 'most unpleasant imaginable touch' on one extreme and 'most pleasant imaginable touch' on the other. Responses were made with the right hand using a response pad system (four-Button Diamond Fiber Optic Response Pad, Current Designs).

16 serial blood samples (of which nine OT and seven cortisol; <70 ml total) were collected from each participant during the session: a pre-run OT and cortisol baseline for each of the two sessions (preceding the T1 anatomical and preceding the resting state scans, respectively); three samples during each run at 1:00 min, 3:30 min (OT only), and 6:30 min; and a final post-session sample outside the scanner (*Figure 1B*). In order to keep the collected blood volume <70 ml, OT but not cortisol sampling was performed at 3:30 min. All blood samples were collected by a nurse and an assistant, who was positioned beside the scanner bore on the left side of the participant. Samples were collected through the indwelling catheter using vacutainer tubes (which rely on the vacuum action of

puncturing of a vial's rubber seal). The two functional runs were separated by ~27 min, allowing OT levels to return to baseline between runs. Samples collected in EDTA-tubes were used for OT analysis and samples collected in serum-gel tubes were used for cortisol analysis. All samples were centrifuged at 4 °C at 10,000 g for 10 min and plasma and serum were then aliquoted and stored at –20 °C until analysis.

After the MR session, the participants evaluated how relaxing they had found partner and stranger touch, how attractive and trustworthy they had found the stranger, and how relaxing the interaction with the nurse had been. All ratings were performed using a visual analog scale (VAS), with –10 as the most negative rating and +10 as the most positive rating.

Occasionally, practical obstacles were encountered, such as coagulation within the catheter or cessation of blood flow from the vein, that resulted in incomplete data series for some participants. We, therefore, sought to maximize analysis for each type of data wherever possible, and so the number of included participants differs between analyzes. See *Supplementary file 6* for details on the participants included and the type of data generated from each participant.

## Hormone analysis

Plasma samples for OT analysis were extracted using acetonitrile precipitation (Merck Millipore: Human Neuropeptide Magnetic Bead Panel 96-Well Plate Assay Cat. # HNPMAG-35K) and OT concentrations were then determined using the Oxytocin ELISA kit (Enzo Life Sciences; sensitivity >15.0 pg/ml, intra-assay precision 10.2–13.3% CV, inter-assay precision 11.8–20.9% CV).

Plasma cortisol levels were determined using the Cortisol Parameter Assay Kit according to the manufacturer's recommendations (R&D Systems, Minneapolis, Minnesota, USA) (sensitivity 0.071 ng/mL, precision 10.4%). Cortisol analysis serum samples were diluted 60 times preceding analysis. Pretreatment steps of the serum samples resulted in a dilution factor of three and the pretreated serum samples required an additional 20-fold dilution in Calibrator Diluent RD5-43.

For both the OT and cortisol analyses, standards and controls were implemented according to manufacturer recommendations. Washing procedures were performed using a Wellwash Microplate Washer (ThermoFisher Scientific, Waltham, Massachusetts, USA) and the absorbance was read using a Multiskan FC Microplate Photometer (ThermoFisher Scientific, Waltham, Massachusetts, USA). The color development of the samples was read for OT at 405 nm (background correction at 571 nm) and for cortisol at 450 nm (background correction at 571 nm). SkanIt Software was used for the creation of standard curves, curve fitting, and calculation of concentrations (ThermoFisher Scientific, Waltham, Massachusetts, USA).

## Statistical analysis of hormone levels

OT levels measured before and during each touch session were entered into a mixed linear model using SPSS version 27 (n=27). Familiarity (partner or stranger) and sample timepoint (pre-run baseline and three samples for each functional run) were included as within-subject factors, whereas order (partner or stranger first) was included as a between-subject factor. Since the Intraclass Correlation Coefficient (ICC) was high (0.73), both fixed and random intercepts were included in the model, and marginal means were estimated through the maximum likelihood method. Post-hoc pairwise comparisons were performed to investigate significant interactions, applying Bonferroni correction to adjust for multiple comparisons.

Student's T-tests were first performed to test for differences in basal cortisol levels depending on the time of day (i.e. sessions beginning at 9:00, 12:00, or 15:00). Time of day did not have a significant effect on the participant's basal cortisol levels and hence all levels were treated equally in the following analysis.

Cortisol levels measured before and during each touch session were entered into a mixed linear model using SPSS version 27 (n=26). As with the oxytocin analysis, familiarity (partner or stranger) and sample timepoint (pre-run baseline and two samples for each functional run) were included as within-subject factors, whereas order (partner or stranger first) was included as a between-subject factor. Since the ICC was high (0.81) both fixed and random intercepts were included in the model and marginal means were estimated through the maximum likelihood method. Post-hoc pairwise comparisons were performed to investigate significant interactions, applying Bonferroni correction to adjust for multiple comparisons.

To investigate whether and how OT and cortisol levels interacted, we defined an additional mixed linear model in SPSS version 27 (n=26). In addition to familiarity (partner or stranger) and sample time-point (pre-run baseline and three samples for each functional run), a within-subject factor for hormone (OT or cortisol) was also included, and order (partner or stranger first) was included as a between-subject factor. Considering that ICC values were high for both OT and cortisol individual models, and we did not assume independence between the two hormone values for each participant, both fixed and random intercepts were included, and marginal means were estimated through the maximum likelihood method. Post-hoc pairwise comparisons were performed to investigate significant interactions, applying Bonferroni correction to adjust for multiple comparisons.

## fMRI data acquisition

fMRI data were acquired using a 3.0 Tesla Siemens scanner (Prisma, Siemens) with a 64-channel head coil. For each functional run, 456 2D T2*-weighted echo-planar images (EPIs) were acquired (repetition time: 901 ms; echo time: 30 ms; slice thickness: 3 mm; no slice gap; matrix size: 64*64; field of view: 488*488 mm$^2$; in-plane voxel resolution: 3 mm$^2$; flip angle: Ernst angle (59°)). Three dummy volumes were acquired before each scan to ensure that data collection started after magnetizations reached a steady state. A high-resolution 3D T1-weighted (MP-RAGE) anatomical image was acquired before the first EPI (repetition time: 2300ms; slice thickness: 0.90 mm; no slice gap; matrix size: 64*64; field of view: 288*288 mm$^2$; voxel resolution: 0.87*0.87*0.90 mm; flip angle: 8°, number of slices: 208).

## fMRI preprocessing and analysis

Preprocessing and statistical analysis of MRI data were performed using Analysis of Functional Neuroimages (AFNI) statistical software (version 19.1.12). Functional data were first de-spiked. Each EPI volume and the T1 were then aligned to the EPI volume with the minimum outlier fraction (using the AFNI outlier definition) to correct for motion. Functional images were warped to the MNI 152 template using a combination of affine and non-linear transformations (*Cox et al., 2017*). Finally, spatial smoothing was applied with a 10 mm full-width at the half-maximum filter. Residual effects of head motion were corrected by including the estimated motion parameters (and their first-order derivatives) as regressors of no interest. A motion censoring threshold of 0.2 mm per TR was implemented in combination with an outlier fraction threshold of 0.1. Volumes violating either of these thresholds were subsequently ignored in the time-series regression. On average, a higher number of volumes was censored for partner arm (14,33±16,81%), than for the other three conditions (partner palm: 9,55±14,18%; stranger arm: 7,78±10,22%; stranger palm: 6,49±8,67%). In line with our approach of maximizing the amount of analyzable data, we decided to not discard additional participants based on the number of censored volumes per condition.

For each participant, whole-brain voxel-wise general linear models (GLM) were created for each of the two runs using 3dDeconvolve. One regressor (convolved with a standard model of the hemodynamic response function, HRF) modeled each of the conditions: partner arm, partner palm, stranger arm, stranger palm. To determine the specific effects of partner and stranger touch regardless of stimulation site, GLMs were created with partner and stranger as regressors. These were used for all analyses, except the linear mixed-effects model with OT covariate (see below). Each regressor modeled 10 s within the 12 s touch interval, beginning 2 s after the onset of touch stimulation and ending with stimulation offset. We also included additional regressors of no interest to model the effects of motor responses during the rating of the touch stimuli.

At the group level, the AFNI program 3dClustSim was used to determine cluster-size thresholds for identifying effects significant at $\alpha$=0.05 family-wise-error (FWE) corrected (within-cluster). Average spatial smoothness estimates, across all participants, used by 3dClustSim were obtained using the 3dFWHMx function with the ACF flag, as per current AFNI recommendations (*Cox et al., 2017*). For each analysis, we report brain areas that were activated at a voxel-wise p-value threshold of p=0.002. Since AFNI outputs a single peak coordinate for each surviving cluster, a custom script was used to extract the coordinates for the first 10 peaks with the highest T scores for each cluster (see accompanying file 'AFNI_10peaks_script').

## Acknowledgements

The authors thank Åsa Axén and Gisela Öhnström for blood sample collection, Kerstin Uvnäs-Moberg, Maria Petersson, Stephanie Preston, and Ellen Lumpkin for valuable discussion, and Paul Hamilton and Irene Perini for assistance with AFNI software. *Funding:* This study was supported by Distinguished Young Investigator grant FYF-2013–687 from the Swedish Research Council to IM.

## Additional information

### Funding

| Funder | Grant reference number | Author |
| --- | --- | --- |
| Vetenskapsrådet | FYF-2013-687 | India Morrison |

The funders had no role in study design, data collection and interpretation, or the decision to submit the work for publication.

### Author contributions

Linda Handlin, Conceptualization, Resources, Data curation, Formal analysis, Investigation, Methodology, Writing – original draft, Writing – review and editing; Giovanni Novembre, Data curation, Formal analysis, Investigation, Visualization, Writing – original draft, Writing – review and editing; Helene Lindholm, Formal analysis, Investigation, Writing – original draft; Robin Kämpe, Formal analysis, Methodology, Writing – review and editing; Elisabeth Paul, Data curation, Formal analysis, Methodology, Writing – review and editing; India Morrison, Conceptualization, Resources, Formal analysis, Supervision, Funding acquisition, Investigation, Methodology, Writing – original draft, Project administration, Writing – review and editing

### Author ORCIDs

Helene Lindholm http://orcid.org/0000-0003-2462-0178
Elisabeth Paul http://orcid.org/0000-0002-0201-273X
India Morrison http://orcid.org/0000-0002-7992-0306

### Ethics

Ethical approval was obtained from the Regional Ethical Review Board in Linköping, Sweden (2015/88-31). All participants gave informed consent in accordance with the Declaration of Helsinki and were compensated at 400 SEK (~45 USD)/h.

### Decision letter and Author response

Decision letter https://doi.org/10.7554/eLife.81197.sa1
Author response https://doi.org/10.7554/eLife.81197.sa2

## Additional files

### Supplementary files

• Supplementary file 1. Paired T-tests for partner vs stranger during each of 2 functional runs (first, second), modeled with linear mixed effects and weighted by the individual change in OT levels as covariates. All contrasts are thresholded at $P<0.002$, cluster-size thresholded at alpha = 0.05 FWE for n=27 complete functional datasets. For each cluster under each contrast heading, size in voxels, location, maximum T score, and MNI coordinates (x, y, z) are given. *=region of interest analysis.

• Supplementary file 2. Paired T-test for partner vs stranger in partner first group, modeled with linear mixed effects and weighted by the individual mean cortisol levels as a covariate. All contrasts are thresholded at $P<0.002$, cluster-size thresholded at alpha = 0.05 FWE for n=18 complete functional datasets. For each cluster under each contrast heading, size, location, maximum T score, and MNI coordinates (x, y, z) are given.

• Supplementary file 3. Regressor created by linear interpolation of serial OT samples, convolved with canonical HRF and modeled with factors toucher (partner, stranger) and order (first or second encounter), at time points 2 and 2.5 min preceding plasma sample collection. All contrasts are

thresholded at P<0.002, cluster-size thresholded at alpha = 0.05 FWE for N=23 complete functional datasets. For each cluster under each contrast heading, size, location, maximum F score, and MNI coordinates (x, y, z) are given.

• Supplementary file 4. Paired T-test for partner vs stranger in partner first group, modeled with linear mixed effects and weighted by individual difference in pleasantness ratings as a covariate. All contrasts are thresholded at P<0.005, cluster-size thresholded at alpha = 0.05 FWE for n=18 complete functional datasets. For each cluster under each contrast heading, size, location, maximum T score, and MNI coordinates (x, y, z) are given.

• Supplementary file 5. Conjunction analyses showing common activations for partner and order factors (partner, stranger, first, second). All contrasts are thresholded at P<0.002, cluster-size thresholded at alpha = 0.05 FWE for n=37 functional datasets. For each cluster under each contrast heading, size in voxels, location, maximum T score, and MNI coordinates (x, y, z) are given. Negative BOLD in boldface.

• Supplementary file 6. Participants included in each analysis based on analyzable data. OT = Oxytocin, CORT = Cortisol, LME = Linear Mixed Effects Model (neuroimaging), Regr.=Regressor (neuroimaging).

• MDAR checklist

• Source code 1. Afni_peak_sourcecode.

## Data availability

Anonymized, unthresholded statistical maps of the fMRI data are available in NeuroVault at: https://identifiers.org/neurovault.collection:13740. Matlab code for the touch paradigm and anonymized hormone values used in the analysis are available at: https://doi.org/10.5061/dryad.zpc866td0. Original plasma sample materials have been consumed during analysis and are therefore inaccessible.

The following datasets were generated:

| Author(s) | Year | Dataset title | Dataset URL | Database and Identifier |
|---|---|---|---|---|
| Morrison I | 2023 | Human endogenous oxytocin and its neural correlates | https://doi.org/10.5061/dryad.zpc866td0 | Dryad Digital Repository, 10.5061/dryad.zpc866td0 |
| Morrison I, Handlin L, Novembre G, Lindholm H, Kämpe R, Paul E | 2023 | Human endogenous oxytocin and its neural correlates show adaptive responses to social touch based on recent social context | https://identifiers.org/neurovault.collection:13740 | NeuroVault, 13740 |

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
