## [Editor Report]

This fundamental work combined naturalistic and neuroscientific methods to demonstrate the context-dependent impact of oxytocin on the brain and behavior. The authors provide compelling evidence that adds significant nuance to our understanding of how social touch is mediated by the brain, which can render people both more and less trusting, depending on conditions. This work will be of broad interest to psychologists and neuroscientists at many levels.

---

## [Decision Letter]

**Decision letter after peer review:**

Thank you for submitting your article "Human endogenous oxytocin and its neural correlates show adaptive responses to social touch based on recent social context" for consideration by *eLife*. Your article has been reviewed by 3 peer reviewers, including Rebecca Shansky as the Reviewing Editor and Reviewer #1, and the evaluation has been overseen by Floris de Lange as the Senior Editor. The following individuals involved in the review of your submission have agreed to reveal their identity: Rebecca Shansky (Reviewer #1); Stephanie Preston (Reviewer #2); Jellina Prinsen (Reviewer #3).

The reviewers have discussed their reviews with one another, and the Reviewing Editor has drafted this to help you prepare a revised submission. Note that all three reviewers were very positive about this manuscript, and recommendations are virtually all to improve clarity so that the broadest readership possible can appreciate the study.

Essential revisions:

*Reviewer #1 (Recommendations for the authors):*

An explanation of the choice to compare arm vs palm touch would be helpful – is there a known difference in how touch to these two body parts is perceived?

*Reviewer #2 (Recommendations for the authors):*

This was an awesome study with a design that was not for the faint of heart. The ability to collect so many concurrent measures, with a study that required three specific people to be present each time, is difficult and rarely attempted. The authors are applauded for this effort.

I so appreciated the study, its results, and the implications for the field that I worry that people won't be able to discern its value readily enough, owing to the complexity.

Experts from this subfield field will already be able to appreciate the study as is, but social bonding and OT are of widespread interest and OT mechanisms are often oversimplified. Thus, I think it's worthwhile to make edits so that the paper can be widely appreciated. To increase the reach, I think the conceptual parts should be more emphasized (e.g.., state the few key hypotheses up front and how they reflect the state of knowledge in the field and dictated the design, what the results were, whether they support or do not support those hypotheses, and what it all means). It's easier for people to follow results when the nature/meaning of the effect is the focus and stats come after or in a table E.g., instead of "there was a three-way interaction of X, Y, and Z," you could say, As predicted, OT levels were sensitive to the context of the interaction, as levels were higher/lower for X partner than Z partner but only when X happened….(demonstrated in the 2x2 interaction STATS, and the main effects of partner STATS, and time STATS--or put those in a table). Here are some places where I noted confusion that could be ameliorated with more direct information:

The use of "afferent tactile stimulation" could be defined at first use.

Sentences like this one (below) were hard to parse because the meaning of others' proposals or frameworks is implicit, which non-experts will not know:

"Proposed functional roles for OT as selectively modulating affiliative social

relationships (34) or maintaining allostatic stability (25) accommodate such differential, context-dependent effects. Nonetheless, a marked gap remains to be bridged between stimulus-driven and context-sensitive frameworks in charting the neural mechanisms of OT effects on social behavior."

Similarly for:

"…stimulus-driven model focused on afferent-subcortical signaling. However, as there is uncertainty surrounding the mechanisms of action of IN-OT and its degree of equivalence to endogenous release (47-50); but see…"

Readers could use more text about the significance of arm versus palm, what was the hypothesis, and what happened.

I am enthusiastic about the inclusion of the partner/stranger conditions and their temporal effects, but more background on how this impacts existing theories and current studies is warranted. E.g., do some not think OT is context sensitive (or do but never showed it)? Dictator-like games with strangers seem to presume relationship doesn't matter (which could explain null effects as shown in recent meta-analyses). Has this effect been demonstrated in rodents but we need to show it in humans? I think some of this is in the text but embedded in citations that myself and others may not know.

The female-first strategy can be defined and called out as a strength, given the predominance of male-only studies, particularly for social bonding. People worry that females will be impacted by cycling hormones and the fact that you did not find this can again be referenced in the discussion to support your approach.

I think readers could use an earlier description of the whole process. E.g., when you start with the IV insertion description, it's not yet clear why they are getting an IV or what the general study design is (this could be at the end of the intro).

Can you add information about how a "good" stranger was selected? Were there criteria/attributes they had to pass? (e.g., not be particularly creepy or attractive from pilot data?)

It wasn't clear until later that the two sessions were during the same visit (many times they are days or weeks apart).

Perhaps say that the "caressing strokes were delivered to the right dorsal arm OR the palm" to make sure it's clear.

Can you clarify why/when in the participants' section there are different amounts of data/people per measure?

Can you state why sometimes only OT is measured and not cort? This could be where their respective typical time course is stated (which comes up later but you need to know during the methods).

Perhaps some of these things are rules of this journal but, e.g., usually, I see stats letters in italics with spaces around the = sign. The tabs on the headers seem strange, but again maybe it's for this journal.

It is difficult for people to unpack interactions, particularly because some involve many levels. It helps to always use directional terms. E.g.,

– At the end of page 10, it says that the post hoc tests revealed a "difference" without saying the direction (which is more).

– The hormone interaction results state at the end that there was an interaction between familiarity and order without the nature of the effect or whether it was hypothesized.

– Similar to the covariate data, when cort results are introduced, most of the paragraph describes the model, and only at the end do we hear about the main effects, but again without specifying the direction or meaning.

– It also says in the discussion that partner compared to stranger touch "covaried" with cortisol without specifying the direction and the Discussion section uses variations of this word in other places. I would not use stats terms (e.g., interaction) in the discussion.

– In general, I would not call it a main effect of familiarity when you can say more directly that people respond X way to their partners over strangers or vice versa.

Palm being worse when it's a stranger first seems not hypothesized. People want to easily distinguish planned comparisons with the theoretical value from results that just emerge from a complex design.

CSI scores have a lot of methods info in the Results section that detracts from knowing the results' relevance or importance. Maybe it is again a journal-specific thing, but it would be easier if the methods/stats were separated so results could focus on what happened and why it was interesting or hypothesized (e.g., if most of the CSI text were above, it could just say in the results that scores were out of 161 and focus on the result).

I was interested in the range of responses to relationship quality. If the range is restricted because they are all good relationships, then it doesn't matter. If there is a wide range, it would be good to know if this impacted responses.

I think Figure 1 could be earlier because I needed it to see the OT levels over time that are reported earlier than the brain results.

The meaning of checking data X seconds before something happens is only clear later. There is more text about this in the discussion (i.e., on the "look back"), but it would be more effective to explain this in the methods/stats and only refer to the outcome and meaning/importance at the end.

What would you say is the conceptual meaning of areas that are conjunctions?

You could bring in real-world relevant issues to this in the discussion such as social anxiety, making friends from strangers, needing human touch from those we love in order to be more receptive to the rest of the world, etc.

Can you state what you think the meaning of the OT stranger second dip is? Like you are receptive to a stranger initially, but then the actual first moments are still difficult with a stranger but you get used to it quicker?

The brain results are hard to digest because there are so many levels of effects, interactions, and lists of areas. Perhaps the area names could be in a table with a more general description of the regions/types of processing and the related psychological phenomena (e.g., "in sensory processing areas, when participants feel more/less trusting/pleasant (Table1)…"). I understand this is hard without reverse inference, but some of these processing-type descriptions are common knowledge.

Does the fact that somatosensory occurs only in conjunction impact your group's (or someone's) theory or the hypothesis that was tested with palm/hand?

In the second half of the discussion paragraph that starts "In recent decades…" (on page 24) there are a lot of acronyms and high-level details that were not really described earlier and the meaning is hard to follow.

I think the discussion should refer back to the females-first approach, how the participant cycle did not impact results (speaking against the usual males-only rationale), and that males should be tested to confirm these results.

There should be a big-picture end sentence or two.

I think this is a fantastic study that can provide us with key information about a topic that is central to people's interests in social-cognitive neuroscience. With some edits, it would be easier for people to discern the importance/rationale of the theories tested, the hypotheses (and whether they were supported or not), and our take-away messages and how they impact our understanding of touch and bonding.

*Reviewer #3 (Recommendations for the authors):*

The positioning of the own study in the oxytocin literature is extensive, but an additional paragraph providing context about what to expect based on the social touch manipulation (i.e. which brain regions are generally activated, how are these modulated by context, etc.) might be enlightening.

In the hypothesis section of the Introduction, it is stated that the paradigm also allowed for investigating whether brain regions with differential responses to touch on the arm as compared to palm skin would show oxytocin-dependent modulation. The authors argue this might be relevant because oxytocin has been proposed to play a role in the signaling of a specific subtype of C afferent nerve (C-tactile), found in hair follicle-containing skin and implicated in affective touch. Yet in the fMRI models including the oxytocin covariate, the factor stimulation site (palm vs. arm) is not included.

Do the authors measure hormonal neuromodulation or neural modulation of hormonal levels? What I'm trying to say here is that multiple times instances of e.g. "endogenous OT modulation of the brain" are used. This seems to imply that plasma OT drives/modulates brain activity (whereas in the Discussion, the authors argue it is probably the other way around). I think other terms are more appropriate here to describe this covarying relationship with max plasma OT levels.

How were the 5 different time windows in the exploratory regression analyses linking earlier BOLD with plasma oxytocin levels selected? Were they chosen arbitrarily, or were they informed by any (experimentally-demonstrated or hypothetical) known time scales of the hormonal (central and peripheral) and neural mechanisms? Do you suspect this covariant pattern in certain subcortical brain areas is a result of central oxytocin binding to its receptors, or are other mechanisms at play? The resulting paragraph in the Discussion might benefit from some additional context.

Although a significant main effect of familiarity was encountered in several brain regions when taking peak plasma oxytocin levels into account, subsequent t-tests showed no activation differences in the BOLD response between partner and stranger touch within the same subjects. Significant interaction maps are thus mainly driven by between-subject effects at different time points. The wording when discussing these results should reflect this (e.g. "Hypothalamus and Raphe nuclei showed BOLD increases during partner touch in the first run." → "Hypothalamus and raphe nuclei showed higher BOLD activations in participants experiencing partner touch vs. participants experiencing stranger touch during the first run.").

Potential to stress the inclusion of female-only participants already in the title and/or abstract more explicitly.

Maintain consistency of time notation (e.g. p. 6: 1m; 3:30 min, 6:30m) and F-test degrees of freedom notation.

Letters used to indicate statistical symbols are always italicized.

Please also include the data acquisition parameters of the anatomical T1 volume adopted for co-registration.

"Residual effects of head motion were corrected by including the estimated motion parameters (and their derivatives) as regressors of no interest" (p. 8). Specify how many derivates were included: first-order, second-order, or full Volterra set.

How many scans were on average censored? Was this number more or less equal during conditions? (i.e. more stress in the stranger-touch condition might have elicited more severe head movements). Was there a cut-off of the minimally required number of TRs?

In the interest of conciseness: similar descriptions of data analysis procedures for OT and cortisol levels and their impact on the BOLD response should be combined in one Data analysis paragraph in the Methods section, including a description of general statistical procedures (e.g. software package used, adopted α cut-off, power calculations, effect sizes, missing data imputation, etc.).

Several abbreviations (STAI-T and STAI-S on p. 13; SVC on p. 15) are not introduced.

The structure of the various functional neuroimaging sections is similar but not fully consistent in terms of heading and subheadings.

For clarity, preference to structure the Results section from "uni-modal" to "multi-modal" results (i.e. from less to more complex); e.g. (0) Self-report questionnaires (these results can also already be included in the Methods section) (1) behavioral rating results, (2) hormonal levels, (3) hormone-independent neural analysis, (4) neural analysis including hormonal covariates/regressors. Also, preference to report the paragraph 'Hormonal cycles and OT levels' (p. 13) under subsection Hormone analysis in Results.

The different subsections of figures 2 and 3 are referred to in the Discussion section of the manuscript but would prefer to also link the different parts of the figure to their corresponding description in the Results.

Inconsistency in error bar depiction between Figure 2A and 3A.

Missing reference for "Cerebrospinal fluid (CSF) has consistently been found to correlate more strongly with brain OT levels than does plasma OT, while CSF and plasma OT measurements have shown weak or no correlation." (p. 24).

Several references occur twice or more in the reference list.

With the all-female sample, the question remains however if and how the current results (and in particular, the time order effects) would generalize to a male population.

[Editors' note: further revisions were suggested prior to acceptance, as described below.]

Thank you for resubmitting your work entitled "Human endogenous oxytocin and its neural correlates show adaptive responses to social touch based on recent social context" for further consideration by *eLife*. Your revised article has been evaluated by Floris de Lange (Senior Editor) and a Reviewing Editor.

The reviewers agree that the manuscript has been greatly improved but there are some minor remaining issues of clarity for Reviewer 2 that need to be addressed, as outlined below. Please submit a revised manuscript that addresses these issues, and we will be able to move forward with publication without another round of peer review.

*Reviewer #2 (Recommendations for the authors):*

I read the revised version of this manuscript.

The paper is clearer; I appreciate the efforts.

The introduction in particular was easier for a non-expert to follow and many results are now described in directional terms, which helped my interpretation. I had a few lingering comments I think could be addressed quickly:

More often the BOLD analyses, now that the directional information is explained, refer to negative relationships, or associations with the less-bonded condition (e.g., more or less for strangers/palms). This is not obvious to a casual reader.

E.g., 3-way interaction in SOG, AG, cuneus for OT decreasing with stranger and decreasing with partner touch (so not just down-regulating). Also greater for stranger in MFG, AG, and more for palm in pre-/post-central gyri. There is a familiarity by OT interaction in AG and MTG that seem like the same areas as those above, but for an opposing result (more for partner > stranger). MTG also increased more with OT increase in stranger first. Also, Partner v Stranger first, dorsal raphe, and hypothalamus are both involved when OT increased for partner > stranger first and decreased in stranger first (the same thematic result).

I assume this does not reflect the subjective response but the up/down regulation? Or bc temporal is so involved with parsing who/what it might increase BOLD to process distinctions about who/what is happening, which has to work harder in less natural situations. Because behavioral/hormonal results are so in line with expectations, and there are so many results, and the figures are labeled as "interactions" without stating the directional info, these finer distinctions will be lost on people

The discussion does say at the top of p19, "noteworthy that the relationship…was an inverse one". Does "more stable OT" mean decreasing? or lower lows and higher highs? Was this pattern similar to the ones in the citations where the directional info is lacking? I think a little more detail in this section would help.

There are still places without directional terms, especially in the discussion. E.g., end of p 15, it says "covaried with cortisol", end of p 18 "mutual influences", "affected by", "Implicated in". 2nd to last para on 19 "dorsal raphe nuclei also covaried with plasma OT", "interactions between OT and familiarity" in AG near the top of p 20. End of p23 "changes" "influenced" "differential".

I get this explains the most general property if gain control produces both directions. But any time a specific result is referred to the direction should be there and then the generality at the beginning and end about how it can produce both decreases and increases will be more clear to people.

Miscellaneous:

Stats reporting has inconsistencies. I would italicize all stats letters and put spacing around = >< signs. If you prefer no spacing (I don't see journal formatting rules), then just make them consistent.

Random errors: E.g., Burbach JP (no initials), says less rather than fewer toward the end of the discussion somewhere, there's an unattached emdash at the beginning of the discussion, and extra space after Tang et al. cite.

I would prefer some NS stats for the linear mixed model with touch and core (e.g., ps >.11).

I don't know if the journal has a max table/figure rule that caused these things below:

Are none of the tables in the main body of the paper? The paper lists them as 1, 2, 3, like they are regular in-text tables but they were all provided as supplements (so should say S2, S3, etc?). I would prefer as a reader to have at least a few key BOLD tables in the paper. The table text could be simplified if you just number the first column (since it already says cluster at the top) and organize by area first then strength to leave all rows blank that repeat the same area. You don't have to do this, of course, but if you put a table in the main text but do not want it to take a lot of room, it would be easy to put a few simplified ones in there. Also, the T is for the t-test? It doesn't clarify. If it is a small stats t some of them would be negative right? Maybe it just refers to the peak voxel, but then IDK what the value is.

The order of the figures presented in the text was confusing. 4B was never mentioned in the text? 2 and 4 are intermixed and not always in order. Can you split up the figures if needed so that only the parts relevant to the text in that place, in order, are present? There is an interesting amygdala graph in Figure 4B, but I didn't see it referred to in the text until the discussion.

Thanks again for a great, interesting paper that I think a lot of people will enjoy. Sorry if I seem pedantic. I just want it to be very accessible to the many people interested in touch, OT, cort, and the brain, so it can have the max reach.

---

## [Author Response]

Essential revisions:Reviewer #1 (Recommendations for the authors):An explanation of the choice to compare arm vs palm touch would be helpful – is there a known difference in how touch to these two body parts is perceived?

Thank you for raising this point. We have now clarified the reasoning and hypotheses surrounding the arm vs palm manipulation throughout different sections of the manuscript. In particular, in the Introduction we provided more explicit explanation about the proposed contribution of CT afferents in arm but not palm touch, and our hypothesis that the associated sensory cortices would not covary with plasma OT (page 5):

“Because both OT and CTs have been implicated in affective touch, it has been proposed that OT may be involved in CT-related neural mechanisms (*5*). Any such link would be supported if brain regions preferentially responding to touch on the CT-rich skin of the arm, compared to the CT-poor skin of the palm, showed specific OT-brain covariation. Alternatively, any cortex-OT modulation for social touch may be independent of any particular peripheral nerve type.”

Following similar comments from all three reviewers, we ran a new linear mixed effects analysis which modelled stimulation site (arm/palm) together with the other regressors, which we had not previously included because the planned analysis had centered on regions of interest defined by an arm vs palm main effect in a general linear model. The results of the new LME confirmed the previous post-hoc t-test results, that is, there was no effect of the OT covariate on BOLD anywhere in the brain, including regions associated with arm vs palm touch.

For the effects of the other included regressors, there were no substantial changes from the results of the previous model. The extent of the regions varied, but the results were essentially the same—ie, the variance accounted for by the presence of the arm-palm regressors did not affect the overall activation map for the other effects.

This new analysis is described in the methods (page 13):

“For each participant, whole-brain voxel-wise general linear models (GLM) were created for each of the two runs using 3dDeconvolve. One regressor (convolved with a standard model of the hemodynamic response function, HRF) modeled each of the conditions: partner arm, partner palm, stranger arm, stranger palm.”

The results are described in the results (page 17) and the image for the interaction between OT and familiarity in Figure 2 has also been updated with the activation map from the new LME.

We also expand on the interpretation of this result in the discussion (page 26):

“However, this experiment provided no supporting evidence for a putative relationship between OT and CT afferent nerve activity associated with affective touch. Primary somatosensory cortex showed selective activation for touch on the palm (Figure 3). Yet no OT-BOLD covariance was observed in arm-specific regions such as the posterior insula/PO (Figure 3), which might have indicated a link between stimulation of CT-rich skin and endogenous OT. In rodent models, oxytocin receptor (OXTR) expression has so far not been identified in dorsal horn neurons of the spinothalamic tract projecting to the specific thalamic pathways putatively shared by CT afferents (*96- 98*). On a subjective level, participants found palm touch from a stranger less pleasant during the initial encounter, potentially reflecting a functional difference in the palm’s prominent role in active sensorimotor exploration (Morrison, 2021) and perhaps potentiation of approach or withdrawal from others’ touch. Further research is therefore needed to explore any functional link between CT afferents and OT.”

Reviewer #2 (Recommendations for the authors):This was an awesome study with a design that was not for the faint of heart. The ability to collect so many concurrent measures, with a study that required three specific people to be present each time, is difficult and rarely attempted. The authors are applauded for this effort.I so appreciated the study, its results, and the implications for the field that I worry that people won't be able to discern its value readily enough, owing to the complexity.Experts from this subfield field will already be able to appreciate the study as is, but social bonding and OT are of widespread interest and OT mechanisms are often oversimplified. Thus, I think it's worthwhile to make edits so that the paper can be widely appreciated. To increase the reach, I think the conceptual parts should be more emphasized (e.g.., state the few key hypotheses up front and how they reflect the state of knowledge in the field and dictated the design, what the results were, whether they support or do not support those hypotheses, and what it all means).

We than the Reviewer for this feedback. We have tried to address this comment by better foregrounding the main messages in several places in the manuscript. In particular, we have highlighted the hypotheses and relevant concepts in the Introduction and tried to improve the explanation of how they are tested in the study.

Namely, in the Introduction we have brought out the tension between approaches to neural OT mechanisms as being sensory-driven vs context sensitive. This is linked to our hypothesis that familiarity should have an effect (partner > stranger), and possibly presentation order (pages 3,4):

“However, the modulatory relationship between OT and the brain is unlikely to be wholly stimulus-driven. It may also be sensitive to aspects of context. For example, the contextual factor of social familiarity plays an important role, with OT-related prosocial behavior frequently limited to familiar individuals (*22-24*). Current physiological state also influences endogenous OT responses (*25,26*). Endogenous OT can mediate prosocial allogrooming behavior in mice (*27*), but OT hormone and OT receptor genotype have also been shown to play roles in antagonistic social behaviors such as defense of offspring (*28*) and aggression (*29,30*). Further, optogenetic manipulation of the same PVN OT neurons in freely-behaving mice can result in either prosocial or antagonistic behavior (*31*). These observations suggest that OT’s role in social behavior is multivalent and situational (*32-34*). Nonetheless, a marked evidential gap remains to be bridged between stimulus-driven and context-sensitive frameworks in charting the neural mechanisms of OT effects on social behavior in humans.

Despite existing indications of context-sensitivity, OT-brain modulation has been little explored for human endogenous OT responses, and how these may be selectively modulated by specific contextual conditions.”

We predicted that touch from a socially familiar person (a romantic partner) would evoke greater endogenous OT changes than touch from an unfamiliar person (a nonthreatening stranger), allowing investigation of the neural responses associated with any such modulation. On the neural level, we expected engagement of hypothalamus and other key regions associated with evolutionarily-conserved circuitry of OT modulation and receptor expression in different species (such as amygdala, medial prefrontal cortex, and cingulate cortex).

We have also tried to emphasize the tension between OT’s role in maintaining stability while also being adaptable to change under certain conditions, which is related to differential responses to partner and stranger, and a predicted interaction between mean cortisol and OT (page 5):

“Beyond these basic questions regarding touch, familiarity, and recent social interaction history, we also investigated any relationship of brain-hormone modulation to stress responses by testing for covariation between OT and peripheral cortisol changes, which can be evoked during a mild, acute stress challenge such as interacting with a socially unfamiliar individual within the novel environment of an fMRI experiment. We predicted lower plasma cortisol levels for the partner compared to the stranger, and an inverse relationship between OT and cortisol measures, during social touch interactions.”

We returned to this in the revised Discussion (pages 22 and 26):

“A single social interaction with a familiar partner is just one instance of many over the course of the relationship. On the other hand, today’s familiar friend was yesterday’s stranger: an interaction with a person one has never met can lay the groundwork for future interactions. Neural and hormonal changes elicited during successive social interactions must therefore not only be able to maintain stability with respect to established social relationships *25*—such as with a romantic partner—but must also be adaptable in the face of new or less certain relationships, such as meeting a new individual. The present findings shed light on the participation of OT in brain-OT covariation during social encounters with both familiar and unfamiliar individuals. They imply that OT and the brain can flexibly coordinate and calibrate responses depending on whom an individual is currently socially interacting with, and with whom the individual has recently interacted.

In everyday life, we humans must navigate a complex and ever-changing social terrain, with some stable elements (for example, established relationships) and other less-stable ones (new or uncertain relationships). This presents a challenge for maintaining stability of existing social bonds on the one hand, yet also establishing and calibrating newer social relationships on the other hand. These findings suggest a role for OT-brain covariation in such adaptive responses. A positive social interaction context (such as a pleasant touch interaction with one’s partner) may selectively bias the system towards a certain neurohormonal response profile, whereas a less certain or less positive social context (such as an unusual interaction with a stranger) may bias it towards a different profile. For example, this could mean that starting the day with a positive social interaction can set up a virtuous circle that perpetuates itself through the day’s social interactions; whereas an uncertain or negative interaction could bias one’s responses towards remaining dampened. Such differential outcomes may potentially influence neural processing and behavior in longer-term social interactions. An important avenue for future research will be to investigate the behavioral effects of these neural and physiological differences, especially with respect to social relationships over time.”

We hope the hypotheses and reasoning around the arm-palm factor are clearer now (see also response to Reviewer 1’s comment).

It's easier for people to follow results when the nature/meaning of the effect is the focus and stats come after or in a table E.g., instead of "there was a three-way interaction of X, Y, and Z," you could say, As predicted, OT levels were sensitive to the context of the interaction, as levels were higher/lower for X partner than Z partner but only when X happened….(demonstrated in the 2x2 interaction STATS, and the main effects of partner STATS, and time STATS--or put those in a table).

We have made changes to phrasing and presentation throughout the Results section, especially in the Discussion, to make it easier for a general reader to parse.

For example, in the Results we now report some main effects and interactions like this (pages 14, 15, 16, 18):

“Partner touch was rated as more pleasant than stranger touch (*F*_1, 33_=30.032, *p*< 0.001) with a main effect of higher ratings for arm (*F*_1, 33_=11.070, *p*=0.002, effect size *f* = 0.7, *partial η2* = 0.33 at power (1-β error probability) = 0.8, *α*=0.05). Participants who received stranger touch first had lower ratings for stranger touch on the palm compared to stranger touch on the arm (*t*=16, *p*=0.007, *d*=1.99), reflected in a significant three-way interaction, indicating influences of both familiarity and order on touch pleasantness ratings for palm (*F*_1, 33_=4.730, *p*=0.037).”

“As predicted, OT levels increased when the partner was the interactant in the first encounter. In addition, when stranger touch was preceded by partner touch OT levels showed a significant dip and recovery during the second touch session. These results were revealed by a three-way interaction between the factors familiarity, order and sample timepoint (*F*_(3, 183.180)_ = 3.034, *p*=0.031).”

“Cortisol showed higher overall levels than OT (a main effect of hormone, *F*_1, 180.593_=68.574, *p*<0.001; OT mean± SEM: partner first: 67.781±6.019, stranger first: 42.959±1.816, p=0.017; cortisol mean± SEM: partner first: 80.346±5.486, stranger first: 101.633±6.306, p=0.040). In addition, OT levels were higher but cortisol levels lower in the partner first condition as compared to stranger first (*F*_1,180.593_=28,751, *p*<0.001). This was revealed by a statistical interaction indicating a mutual influence between familiarity and order (*F*_1, 178.355_=10.565, *p*=0.001; also found in the two previous individual models).”

“BOLD was greater for stranger than partner (a main effect of familiarity) in bilateral middle frontal gyrus (MFG) and right AG, among other regions. BOLD was greater for palm than arm (a main effect of stimulation site) in left postcentral gyrus (PoCG) and bilateral precentral gyrus (PrCG).”

“In parietotemporal clusters, the higher an individual’s BOLD, the higher her OT levels when receiving partner touch in the second run, as compared to those receiving stranger touch in the second run (in whom this OT-BOLD covariation was less positive). These parietotemporal clusters were also seen in the interaction maps in which OT interacted with familiarity (right AG, right TP; Figure 2) and order (right MTG). Additional clusters were revealed in right superior temporal gyrus (STG), among other activations.”

In the Discussion we have revised the text to lead with the interpretation of the statistical effects, for example (pages 22, 24, 25):

“The most general finding was that touch-mediated social interactions in human females elicited endogenous OT and brain responses in a covariant manner. Beyond this, OT and neural changes were modulated by the familiarity of the person delivering touch, as well as the recent history of social interaction. The effect of these contextual factors on within-subject endogenous OT changes manifested in a mutual influence between the familiarity of the social interactant (partner or stranger) and the order of his presentation over two immediately successive social interactions (partner then stranger, or stranger then partner). This influence was driven by a greater increase in plasma OT responses for the stranger following partner touch, whereas there was no corresponding increase for partner touch following stranger touch (Figure 2A).”

“When the stranger touched first, BOLD in both hypothalamus and dorsal raphe was greater the lower mean OT was across individuals (Figure 3D), again suggestive of dynamic co-modulation.”

“Here, mean plasma cortisol levels were higher for stranger than for partner during the first encounter (Figure 3A) and decreased as mean OT levels increased, indicating an inverse relationship between OT and cortisol in peripheral circulation during this experiment.”

Here are some places where I noted confusion that could be ameliorated with more direct information:The use of "afferent tactile stimulation" could be defined at first use.

We have now changed this.

Sentences like this one (below) were hard to parse because the meaning of others' proposals or frameworks is implicit, which non-experts will not know:"Proposed functional roles for OT as selectively modulating affiliative socialrelationships (34) or maintaining allostatic stability (25) accommodate such differential, context-dependent effects. Nonetheless, a marked gap remains to be bridged between stimulus-driven and context-sensitive frameworks in charting the neural mechanisms of OT effects on social behavior."

Thank you for pointing out these dense spots in the text. We have tried to unpack the references throughout the manuscript. This passage has been taken out.

Similarly for:"…stimulus-driven model focused on afferent-subcortical signaling. However, as there is uncertainty surrounding the mechanisms of action of IN-OT and its degree of equivalence to endogenous release (47-50); but see…"

We have now clarified this in the introduction (page 3):

There is uncertainty surrounding the mechanisms of action of IN-OT and its degree of equivalence to endogenous release, chiefly regarding the questions of whether the molecule crosses the blood-brain barrier, and how peripheral effects can be disentangled from central effects (*47-50*; but see *51*).

Readers could use more text about the significance of arm versus palm, what was the hypothesis, and what happened.

We have revised the introduction and discussion to clarify this. We also reanalyzed the data with a new model that included these regressors alongside the others—please see the response to Reviewer 1.

I am enthusiastic about the inclusion of the partner/stranger conditions and their temporal effects, but more background on how this impacts existing theories and current studies is warranted. E.g., do some not think OT is context sensitive (or do but never showed it)? Dictator-like games with strangers seem to presume relationship doesn't matter (which could explain null effects as shown in recent meta-analyses). Has this effect been demonstrated in rodents but we need to show it in humans? I think some of this is in the text but embedded in citations that myself and others may not know.

Thank you. We regard one of the interesting advances of these results to be their indication that relationship does matter, implying that specific OT-related effects in humans may elude capture unless factors like familiarity, social relationship, current context, etc, are taken into account. To our knowledge no one has directly demonstrated this so far for human endogenous OT, though it has been proposed based on less direct data (eg Quintana et al. 2021, Bartz et al. 2011), for example familiarity effects in rodents. One recent rodent study used optogenetic stimulation of OT-expressing neurons in PVN to demonstrate that mice can exhibit either prosocial or antagonistic behavior depending on the situation (Yu et al., 2022).

We have added this text in the Discussion (page 25):

“Taken together, these selective hormone-brain changes support the view that endogenous OT’s role in human social interaction is heavily modulated by contextual factors (*26*), and provides further evidence that this role can involve modulation in a positive or a negative direction, depending on the situation (*28-30*). For example, in prairie voles, higher levels of endogenous OT can mediate prosocial grooming of stressed others (*24*), but optogenetic manipulation of the same PVN OT neurons in freely-behaving mice can result in either prosocial or antagonistic behavior (*28*). OT and OT receptor genotype have also been shown to play a role in antagonistic social behaviors such as defense of offspring in rats (*25*) and aggression in rodents and humans (*26,27*).”

The female-first strategy can be defined and called out as a strength, given the predominance of male-only studies, particularly for social bonding. People worry that females will be impacted by cycling hormones and the fact that you did not find this can again be referenced in the discussion to support your approach.

We have revised the text in the introduction and discussion to try to bring this point out more saliently.

We have also made the female-first strategy more prominent (pages 4 and 27):

“In this study, we examined whether social interactions involving touch can evoke endogenous changes in plasma OT in human females, and whether this would be modulated by interacting with a socially familiar individual. Given the preponderance of OT studies in male populations, we also took a “female-first” strategy (55) by testing hormone and brain responses in a female population.

Most human studies manipulating OT (usually via nasal administration) have been performed in male populations. In contrast, the present study used a “female-first” strategy which moves to redress this imbalance (6); likewise recent research in rats has focused on OT-touch mechanisms in female samples (*1, 14*). Here we found no effect of cycle phase on evoked OT changes. An important question for future research is whether, and to what extent, these results in human females generalize to males, especially with regard to any familiarity-dependent bias in endogenous OT. Another potential sex difference may lie in the relationships between OT, cortisol, and their covariation in mPFC.”

I think readers could use an earlier description of the whole process. E.g., when you start with the IV insertion description, it's not yet clear why they are getting an IV or what the general study design is (this could be at the end of the intro).

We have added more description of the basic study design at the end of the Introduction (page 5):

“Crucially, we also explored whether these OT-brain modulation changes would be modulated by participants’ very recent social interaction history with familiar or unfamiliar others. Each participant was caressed by both their partner and an unfamiliar yet unthreatening stranger over two successive parts of the same experimental session, while a total of eight plasma OT samples and six plasma cortisol samples per participant were collected over the session. The presentation order of partner or stranger across the two successive touch interactions during the experiment was counterbalanced: the stranger’s touch could either precede the partner’s touch or come after it.”

Can you add information about how a "good" stranger was selected? Were there criteria/attributes they had to pass? (e.g., not be particularly creepy or attractive from pilot data?)

When we designed the experiment we wanted to include a check to exclude the possibility that the male stranger was perceived by the participants either too positively or too negatively, which might have introduced systematic biases in the data. We therefore collected post-manipulation ratings of attractiveness and trustworthiness (described in the methods section). These ratings fell to neither extreme (Results page 14).

Although there was no specific selection process for the stranger in this study, we had performed pilot experiments (2 behavioral/hormone, 1 procedural for fMRI) in which male lab colleagues acted as the stranger. These indicated that these males were not perceived as too negative or positive, so they (in most cases the same individual) also took the stranger role in the main experiment.

It wasn't clear until later that the two sessions were during the same visit (many times they are days or weeks apart).

In the methods section we have now added the following sentence to make this clearer: “The participants performed both runs during a single visit to the lab, i.e. they received touch from both partner and stranger during the same visit.” (page 9)

Perhaps say that the "caressing strokes were delivered to the right dorsal arm OR the palm" to make sure it's clear.

It has now been clarified in the methods section (page 9) that the touch was applied to either arm or palm during each 12-s touch stimulation trial.

Can you clarify why/when in the participants' section there are different amounts of data/people per measure?

We collected data as described for all participants. However, sometimes we encountered technological and physiological difficulties, such as cessation of blood flow through the catheter due to coagulation or vessel constriction, or scanner problems, etc. that resulted in incomplete data series for some participants. Where participants had missing data, we maximized the analysis for each type of data (OT, cortisol, fMRI) and therefore the number of included participants sometimes differ between analyses.

This information is now added in the methods section (page 10):

“Occasionally, practical malfunctions were encountered, such as coagulation within the catheter or cessation of blood flow from the vein, that resulted in incomplete data series for some participants. We therefore sought to maximize analysis for each type of data wherever possible, and so the number of included participants differs between analyzes. See Table 1 for details on participants included and type of data generated from each participant.”

Can you state why sometimes only OT is measured and not cort? This could be where their respective typical time course is stated (which comes up later but you need to know during the methods).

Oxytocin is a more rapidly-responding hormone than cortisol, which tends to change at a slower timescale. Therefore we collected more frequent measurements of oxytocin during the touch sessions, compared to cortisol. In addition, we wanted to keep the collected blood volume to < 70ml so only OT, but not cortisol, samples were collected at 3:30 min in each functional run (visualized in Figure 1). This is now added to the methods section (page 9):

In order to keep the collected blood volume < 70 ml, OT but not cortisol sampling was performed at 3:30 min.

Perhaps some of these things are rules of this journal but, e.g., usually, I see stats letters in italics with spaces around the = sign. The tabs on the headers seem strange, but again maybe it's for this journal.

We have gone through the manuscript and made changes so that the formatting of the text is consistent.

It is difficult for people to unpack interactions, particularly because some involve many levels. It helps to always use directional terms. E.g.,– At the end of page 10, it says that the post hoc tests revealed a "difference" without saying the direction (which is more).– The hormone interaction results state at the end that there was an interaction between familiarity and order without the nature of the effect or whether it was hypothesized.– Similar to the covariate data, when cort results are introduced, most of the paragraph describes the model, and only at the end do we hear about the main effects, but again without specifying the direction or meaning.– It also says in the discussion that partner compared to stranger touch "covaried" with cortisol without specifying the direction and the Discussion section uses variations of this word in other places. I would not use stats terms (e.g., interaction) in the discussion.– In general, I would not call it a main effect of familiarity when you can say more directly that people respond X way to their partners over strangers or vice versa.

Thank you for pointing out these specific places. We have now changed the wordings when we present the results in an attempt to make it clearer what the statistical analyses are showing, including directions and interactions.

Palm being worse when it's a stranger first seems not hypothesized. People want to easily distinguish planned comparisons with the theoretical value from results that just emerge from a complex design.

This is a very good point. We have attempted to clarify the hypothesis for the arm vs palm comparison in the Intro (see also the reply to Reviewer 1’s comment).

Regarding the palm being rated less pleasant specifically in the stranger first condition, we neither predicted this nor are we confident about its interpretation. We have now made this explicit in the discussion (page 26):

On a subjective level, participants found palm touch from a stranger less pleasant during the initial encounter, which was not predicted. This potentially reflects a functional difference in the palm’s prominent role in active sensorimotor exploration, and perhaps potentiation of approach or withdrawal from others’ touch. Further research is therefore needed to explore any functional link between CT afferents and OT.

CSI scores have a lot of methods info in the Results section that detracts from knowing the results' relevance or importance. Maybe it is again a journal-specific thing, but it would be easier if the methods/stats were separated so results could focus on what happened and why it was interesting or hypothesized (e.g., if most of the CSI text were above, it could just say in the results that scores were out of 161 and focus on the result).

In an attempt to make it clearer we have now changed the text so that the description of the CSI and its statistical analysis is now presented under the Methods section instead and only the results from the inventory are now in the Results section.

I was interested in the range of responses to relationship quality. If the range is restricted because they are all good relationships, then it doesn't matter. If there is a wide range, it would be good to know if this impacted responses.

The range is now displayed through the standard deviation values reported in parentheses (page 13):

CSI scores indicated that both female and male participants were satisfied with their relationships (females: mean=139.8, sd=17.2, males: mean=139.6, sd=15.2) and the participants’ assessments of relationship quality correlated with their partners’ (*r*=0.57, *p*=0.0003).

These values indicate that the range is restricted, which supports the finding that the participants considered their relationships to be good. However, it should be noted that there may have been a selection bias in the sample, assuming that couples who get along well are more likely to volunteer for this kind of experiment.

I think Figure 1 could be earlier because I needed it to see the OT levels over time that are reported earlier than the brain results.

Figure 1 is now after “Procedure” subsection in Methods. We have also made changes to the figures. Figure 2 now presents both OT and cortisol and basic BOLD-hormone covariance results. Figure 3 presents the exploratory temporal OT regressor results, the non-hormone-dependent results, and ITG covariation with pleasantness (also with OT and order, and OT and familiarity).

Previously, the ITG results for touch pleasantness covariation were presented in the figure and figure legend but not in the body of the text. We have fixed this oversight (page 21):

T-test with touch pleasantness covariate

To discover regions in which BOLD activation covaried with changes in touch pleasantness ratings, we performed t-tests between partner and stranger for each presentation order (partner first or stranger first) with the difference in pleasantness ratings (partner minus stranger) as covariate of interest. There was no resulting activation in the stranger first group. In the partner first group, a cluster in ITG was revealed, which overlapped with the clusters in which OT interacted with familiarity and order, respectively (Figure 3D, Table S7). Here, individuals that showed higher BOLD activation for partner compared to stranger also showed the largest difference in ratings between partner and stranger touch. When the stranger delivered touch in the second encounter, ITG activation increased with touch pleasantness during the run (r = 0.65, p = 0.003; Figure 3D).

The meaning of checking data X seconds before something happens is only clear later. There is more text about this in the discussion (i.e., on the "look back"), but it would be more effective to explain this in the methods/stats and only refer to the outcome and meaning/importance at the end.

We have changed the text in both methods and discussion, and hope it is clearer now. Reviewer 3 also raised a related point about directionality. We have therefore also tried to clarify that we assumed a central-to-peripheral direction of influence in creating this regressor. However, we believe that the full causal picture is likely to involve a modulatory loop (or even loops) in which peripheral and central changes influence one another.

Results (page 19):

“We assumed that any central-to-peripheral effects of OT release would be detectable retrospectively by modeling the plasma OT sample points “backwards”, in order to search for any BOLD activity which both preceded and tracked the observed pattern of OT changes. Points between the multiple samples were linearly interpolated and the resulting function was convolved with the canonical hemodynamic response function (HRF). N=23 participants had complete data series for both functional runs (Table XX). For this “backward-looking” regressor, we explored time lags of 1, 1.5, 2, 2.5 and 3 min to capture potential touch-evoked central modulation corresponding to the peripheral changes in plasma OT observed after these various delays, assuming that central activity preceded peripheral OT changes.”

Discussion (page 24):

“Central effects of IN-OT have consistently been found ~45 min post-administration (*75-77*), but there is less direct evidence about the timecourse of central endogenous OT release into the periphery in humans. We therefore developed an exploratory regressor based on the serial pattern of individuals’ OT levels. Assuming a mechanism in which plasma OT changes were affected by central release in the brain and thus came after it in time, this regressor allowed us to look “backwards” from the temporal pattern of the plasma OT sample series to any preceding hemodynamic activation that tracked with this pattern. The pattern-covariant engagement of the precuneus at 2.5 minutes preceding sampling, and of precuneus, retrosplenial cortex, and mPFC at 2 minutes, is within the frame of the half-life of OT in blood (*78*) and may reflect events surrounding central OT release (*15*). Although the present results lack sufficient temporal and causal resolution to address this, it is possible that any descending OT influence from the brain to the periphery instates a “reafferent loop” in which central-to-peripheral changes can, in turn, influence incoming sensory information, perhaps at the level of spinal and/or brainstem mechanisms.”

What would you say is the conceptual meaning of areas that are conjunctions?

The conjunctions reflect regions that are engaged by the touch manipulation across conditions, regardless of familiarity, order, hormone levels, or stimulation site. The activation pattern is consistent with somatosensory stimulation, and we infer that it is relatively robust to the more context-dependent variables in the paradigm. The presence of these “general” touch activations also complements the lack of covariation with OT levels here. We interpret this as further supporting the idea that these context-dependent variables do not directly affect processing in somatosensory cortices or the other regions in the conjunction.

You could bring in real-world relevant issues to this in the discussion such as social anxiety, making friends from strangers, needing human touch from those we love in order to be more receptive to the rest of the world, etc.

Thank you for this good suggestion. We have added some text to the “limitations and future directions” section of the discussion (pages 27-28) speculating that touch/interaction with a loved one might make one more receptive, or at least help to buffer one against any negative/uncertain circumstances in the short term:

“In everyday life, we humans must navigate a complex and ever-changing social terrain, with some stable elements (for example, established relationships) and other less-stable ones (new or uncertain relationships). This presents a challenge for maintaining stability of existing social bonds on the one hand, yet also establishing and calibrating newer social relationships on the other hand. These findings suggest a role for OT-brain covariation in such adaptive responses. A positive social interaction context (such as a pleasant touch interaction with one’s partner) may selectively bias the system towards a certain short-term neurohormonal response profile, whereas a less certain or less positive social context (such as an unusual interaction with a stranger) may bias it towards a different profile. For example, this could mean that starting the day with a positive social interaction can set up a virtuous circle that perpetuates itself through the day’s social interactions; whereas an uncertain or negative interaction could bias one’s responses towards remaining dampened. Such differential outcomes may potentially influence neural processing and behavior in longer-term social interactions. An important avenue for future research will be to investigate the behavioral effects of these neural and physiological differences, especially with respect to social relationships over time.”

Can you state what you think the meaning of the OT stranger second dip is? Like you are receptive to a stranger initially, but then the actual first moments are still difficult with a stranger but you get used to it quicker?

The dip might reflect an initial destabilization (decrease from baseline) with the onset of the stranger, but which recovers to above-baseline levels over time. The observation that this happened for stranger only when preceded by partner may be attributed to a gain-setting effect of context.

We have added more detail of this interpretation in the results (page 15):

“As predicted, OT levels increased when the partner was the interactant in the first encounter. In addition, when stranger touch was preceded by partner touch OT levels showed a significant dip and recovery during the second touch session. These results were revealed by a three-way interaction between the factors familiarity, order and sample timepoint (*F*_(3, 183.180)_ = 3.034, *p*=0.031). A significant increase between the first and middle samples in the functional run during stranger touch in the partner first group only (*p*=0.027) drove the contribution of timepoint to the three-way interaction.”

and Discussion (page 23):

“Social familiarity and presentation order also influenced the timecourse of the eight plasma samples collected during the experimental session, with this influence driven by OT responses to stranger touch following partner touch. The OT increase for stranger touch in this condition did not show a stable rise, but rather dipped to below-baseline levels across participants before recovering to above-baseline levels by the end of the social encounter. A tentative interpretation for this pattern is that the initial partner encounter may have introduced a bias for OT increase during the subsequent stranger encounter, though not as a sustained carryover from the preceding partner interaction. In contrast, plasma OT remained at baseline levels when the experimental session began with an encounter with an unfamiliar stranger. The recovery of OT in the stranger-second condition could thus reflect a facilitation of underlying endogenous release mechanisms following the prior partner interaction. Overall, these endogenous OT changes may reflect mechanisms that selectively bias the way social stimuli are processed in the central nervous system during social interactions, with high dependence on contextual information.”

The brain results are hard to digest because there are so many levels of effects, interactions, and lists of areas. Perhaps the area names could be in a table with a more general description of the regions/types of processing and the related psychological phenomena (e.g., "in sensory processing areas, when participants feel more/less trusting/pleasant (Table1)…"). I understand this is hard without reverse inference, but some of these processing-type descriptions are common knowledge.

We have rephrased the brain results in various places to try to make the levels more clear. We hope it has become easier to parse with the restructuring of the Methods and Results sections and the new model which includes arm/palm regressors, and also the revised tables. Also, we have moved some “listed” brain areas to the tables. We are wary of reverse inference, as you pointed out, but for certain regions we are more confident in invoking associated functions, such as medial prefrontal cortex for stress regulation, and temporo-parietal pathways which are implicated in a broad range of “social” tasks.

Does the fact that somatosensory occurs only in conjunction impact your group's (or someone's) theory or the hypothesis that was tested with palm/hand?

We have tried to make the arm-palm hypothesis and the potential links between arm skin stimulation and CT afferent activity more explicit in both the intro and the discussion. (See also the response to Reviewer 1’s comment.)

In the second half of the discussion paragraph that starts "In recent decades…" (on page 24) there are a lot of acronyms and high-level details that were not really described earlier and the meaning is hard to follow.

Thank you for pointing this out. This paragraph has been rewritten and hopefully purged of jargon. We hope it is clearer now.

I think the discussion should refer back to the females-first approach, how the participant cycle did not impact results (speaking against the usual males-only rationale), and that males should be tested to confirm these results.

Following this suggestion and a similar one from Reviewer 3, we have added text to the Discussion (see also the response to a related comment above).

There should be a big-picture end sentence or two.

We have now added this text to the Discussion following a “real-world” scenario (page 28):

“Conclusions

These findings offer a methodological and conceptual bridge between stimulus-driven and context-sensitive frameworks of endogenous OT modulation of the brain during social interactions. Touch-mediated social interactions evoked changes in endogenous OT, indicating the importance of the stimulus. Yet these responses were nevertheless influenced by specific features of social context, with plasma of OT levels showing differential responses depending on the familiarity of the interacting person and the recent history of interaction. Such adaptive responses could reflect a gain-control-like role for OT-brain neuromodulation, comparable to a dimmer switch, which can effectively preserve stability with respect to established social relationships while also allowing for change in new ones (possibly via increases or decreases in inhibitory influence). Across successive social encounters, such modulatory mechanisms may calibrate neural and behavioral receptivity, whether mediated by touch or another channel such as vision or speech. Network hubs in parietotemporal pathways, alongside precuneus and retrosplenial cortex, may be key for turning the “dimmer” of OT-brain processing up or down depending on past and current social context.”

I think this is a fantastic study that can provide us with key information about a topic that is central to people's interests in social-cognitive neuroscience. With some edits, it would be easier for people to discern the importance/rationale of the theories tested, the hypotheses (and whether they were supported or not), and our take-away messages and how they impact our understanding of touch and bonding.

Thank you!

Reviewer #3 (Recommendations for the authors):The positioning of the own study in the oxytocin literature is extensive, but an additional paragraph providing context about what to expect based on the social touch manipulation (i.e. which brain regions are generally activated, how are these modulated by context, etc.) might be enlightening.

We have now clarified the main hypotheses and concepts in the Introduction (see also the reply to a similar comment from Reviewer 2). Regarding the specific expectations of brain activations and modulations, there has not been very much previous research on human endogenous OT to go on, but we hope the revised text in the introduction makes our predictions more clear.

In the hypothesis section of the Introduction, it is stated that the paradigm also allowed for investigating whether brain regions with differential responses to touch on the arm as compared to palm skin would show oxytocin-dependent modulation. The authors argue this might be relevant because oxytocin has been proposed to play a role in the signaling of a specific subtype of C afferent nerve (C-tactile), found in hair follicle-containing skin and implicated in affective touch. Yet in the fMRI models including the oxytocin covariate, the factor stimulation site (palm vs. arm) is not included.

Thank you for raising this important point. We had originally planned the arm-palm manipulation as a way of defining ROIs to check for OT covariance in posterior insula/parietal operculum (arm) and primary somatosensory cortices (palm). But we very much agree that including this factor in the full model is preferable, so we have re-run a linear mixed effects model accordingly (see also response to Reviewer 1’s comment). We replaced the previous linear mixed model (LME) Order *Familiarity with OT covariate and the GLM Familiarity* Site with a bigger LME model Order *Familiarity* Site with OT covariate. In this way we could test the covariation between the Site of stimulation and OT changes and the interaction with the other experimental factors (Order and Familiarity).

No covariance with OT was seen, consistent with the previous inference based on the absence of arm/palm regions in t-tests. The whole-brain results were not substantially different from those of the previous model. The methods, results, tables, and Figure 2 have been updated to reflect the new LME.

Do the authors measure hormonal neuromodulation or neural modulation of hormonal levels? What I'm trying to say here is that multiple times instances of e.g. "endogenous OT modulation of the brain" are used. This seems to imply that plasma OT drives/modulates brain activity (whereas in the Discussion, the authors argue it is probably the other way around). I think other terms are more appropriate here to describe this covarying relationship with max plasma OT levels.

This is a great point, and we agree. We have removed language implying a direction of modulation throughout the manuscript, and replaced it with “brain-OT covariance” or “comodulation” and similar expressions.

How were the 5 different time windows in the exploratory regression analyses linking earlier BOLD with plasma oxytocin levels selected? Were they chosen arbitrarily, or were they informed by any (experimentally-demonstrated or hypothetical) known time scales of the hormonal (central and peripheral) and neural mechanisms? Do you suspect this covariant pattern in certain subcortical brain areas is a result of central oxytocin binding to its receptors, or are other mechanisms at play? The resulting paragraph in the Discussion might benefit from some additional context.

The temporal OT regressor was not based on any specific experimentally-demonstrated time scales of OT release in any organism. The time lags were selected with the aim of exploring the most possible time lags allowed by the number of sample points within the total time window of 5.5 minutes (the first sample in the run was collected and the last at 6.5 minutes). Thus the lags were selected partly arbitrarily and partly with an eye to the total timeframe of the 7-min functional runs. We thought that including these 5 lags was a good compromise between the exploratory nature of the analysis and the reliability of the methodological approach. In a general sense, the creation of the regressor as “backward-looking” was informed by evidence from animal models for central-to-peripheral effects from hypothalamus to pituitary to bloodstream. We assumed this directionality in the analysis. The text in methods, results, and the discussion has been revised to make this assumption explicit as such.

See also the reply to a related comment from Reviewer 2.

We did not interpret the BOLD in terms of any specific cell-level mechanisms such as receptor binding, but we would speculate that these may be involved, whether directly or indirectly. We think it is very likely that the BOLD reflects some degree of mediated effect between underlying cellular mechanisms and larger-scale network activity (and possibly even peripheral feedback). Unfortunately, these issues can’t be addressed with this dataset, but the results do suggest avenues for stronger tests in the future. For example, since resting-state evidence of precuneus and retrosplenial engagement comes from IN-OT administration studies, manipulations combining contextual factors, endogenous OT, and exogenous OT might be fruitful.

Although a significant main effect of familiarity was encountered in several brain regions when taking peak plasma oxytocin levels into account, subsequent t-tests showed no activation differences in the BOLD response between partner and stranger touch within the same subjects. Significant interaction maps are thus mainly driven by between-subject effects at different time points. The wording when discussing these results should reflect this (e.g. "Hypothalamus and Raphe nuclei showed BOLD increases during partner touch in the first run." → "Hypothalamus and raphe nuclei showed higher BOLD activations in participants experiencing partner touch vs. participants experiencing stranger touch during the first run.").

We are grateful to the Reviewer for making us aware of the lack of clarity here. We agree it is very important to make clear that, as in most covariation effects in fMRI studies, the effects are driven by interindividual differences for a given covariant relationship, rather than the within-subject BOLD response increasing or decreasing.

We have now rephrased the results for the Partner First vs. Stranger First and Partner Second vs. Stranger Second t-test, making clear that the observed effects in the t-tests arise from between-group comparisons. For example (page 18):

“Partner Second vs. Stranger Second: In parietotemporal clusters, the higher an individual’s BOLD, the higher her OT levels when receiving partner touch in the second run, as compared to those receiving stranger touch in the second run (in whom this OT-BOLD covariation was less positive). These parietotemporal clusters were also seen in the interaction maps in which OT interacted with familiarity (right AG, right TP; Figure 2) and order (right MTG). Additional clusters were revealed in right superior temporal gyrus (STG), among other activations.”

Potential to stress the inclusion of female-only participants already in the title and/or abstract more explicitly.

We have now revised the Introduction and Discussion accordingly. See also the response to a similar comment from Reviewer 2.

Maintain consistency of time notation (e.g. p. 6: 1m; 3:30 min, 6:30m) and F-test degrees of freedom notation.

This has been done.

Letters used to indicate statistical symbols are always italicized.

This has been done.

Please also include the data acquisition parameters of the anatomical T1 volume adopted for co-registration.

We have now added the following sentence:

“A high-resolution 3D T1-weighted (MP-RAGE) anatomical image was acquire before the first EPI (repetition time: 2300 ms;; slice thickness: 0.90 mm; no slice gap; matrix size: 64 × 64; field of view: 288 × 288 mm2; voxel resolution: 0.87*0.87*0.90 mm; flip angle: 8°, number of slices: 208).”

"Residual effects of head motion were corrected by including the estimated motion parameters (and their derivatives) as regressors of no interest" (p. 8). Specify how many derivates were included: first-order, second-order, or full Volterra set.

Thanks for pointing out the missing information. We have made clear now that we included the first-order derivatives.

How many scans were on average censored? Was this number more or less equal during conditions? (i.e. more stress in the stranger-touch condition might have elicited more severe head movements). Was there a cut-off of the minimally required number of TRs?

We have now added this information to the manuscript.

“On average, a higher number of volumes was censored for partner arm (14,33±16,81%), than for the other three conditions (partner palm: 9,55±14,18%; stranger arm: 7,78±10,22%; stranger palm: 6,49±8,67%). In line with our approach of maximizing the amount of analyzable data, we decided to not discard additional participants based on the number of censored volumes per condition.”

In the interest of conciseness: similar descriptions of data analysis procedures for OT and cortisol levels and their impact on the BOLD response should be combined in one Data analysis paragraph in the Methods section, including a description of general statistical procedures (e.g. software package used, adopted α cut-off, power calculations, effect sizes, missing data imputation, etc.).

This has been done.

Several abbreviations (STAI-T and STAI-S on p. 13; SVC on p. 15) are not introduced.

This has been corrected. Please see response to reviewer #2.

The structure of the various functional neuroimaging sections is similar but not fully consistent in terms of heading and subheadings.

Thank you for helping us with the structure of the manuscript. We have now fixed the headings and subheadings of the fMRI sections.

For clarity, preference to structure the Results section from "uni-modal" to "multi-modal" results (i.e. from less to more complex); e.g. (0) Self-report questionnaires (these results can also already be included in the Methods section) (1) behavioral rating results, (2) hormonal levels, (3) hormone-independent neural analysis, (4) neural analysis including hormonal covariates/regressors. Also, preference to report the paragraph 'Hormonal cycles and OT levels' (p. 13) under subsection Hormone analysis in Results.

Thank you for this suggestion. The Results section has now been re-structured. Specifically, it now follows this structure: behavior (questionnaires and ratings), hormones (OT and cortisol), functional neuroimaging (LME, t-tests, OT regressor, ditto for cortisol), hormone-independent analysis.

The different subsections of figures 2 and 3 are referred to in the Discussion section of the manuscript but would prefer to also link the different parts of the figure to their corresponding description in the Results.Inconsistency in error bar depiction between Figure 2A and 3A.

These have now been fixed.

Missing reference for "Cerebrospinal fluid (CSF) has consistently been found to correlate more strongly with brain OT levels than does plasma OT, while CSF and plasma OT measurements have shown weak or no correlation." (p. 24).

Thank you for pointing this out, we have now added this reference:

Discussions and perspectives regarding oxytocin as a biomarker in human investigations.

Several references occur twice or more in the reference list.

We have now fixed this.

With the all-female sample, the question remains however if and how the current results (and in particular, the time order effects) would generalize to a male population.

We wonder the same thing, and have speculative ideas which we have now added to the Future Directions section of the discussion (page 27):

“Most human studies manipulating OT (usually via nasal administration) have been performed in male populations. In contrast, the present study used a “female-first” strategy which moves to redress this imbalance (*55*); likewise recent research in rodents has focused on OT-touch mechanisms in female samples (*1, 28*). Here we found no effect of cycle phase on evoked OT changes. An important question for future research is whether, and to what extent, these results in human females generalize to males, especially with regard to any familiarity-dependent bias in endogenous OT. Another potential sex difference may lie in the relationships between OT, cortisol, and their covariation in mPFC.”

[Editors' note: further revisions were suggested prior to acceptance, as described below.]

The reviewers agree that the manuscript has been greatly improved but there are some minor remaining issues of clarity for Reviewer 2 that need to be addressed, as outlined below. Please submit a revised manuscript that addresses these issues, and we will be able to move forward with publication without another round of peer review.Reviewer #2 (Recommendations for the authors):I read the revised version of this manuscript.The paper is clearer; I appreciate the efforts.The introduction in particular was easier for a non-expert to follow and many results are now described in directional terms, which helped my interpretation. I had a few lingering comments I think could be addressed quickly:More often the BOLD analyses, now that the directional information is explained, refer to negative relationships, or associations with the less-bonded condition (e.g., more or less for strangers/palms). This is not obvious to a casual reader.E.g., 3-way interaction in SOG, AG, cuneus for OT decreasing with stranger and decreasing with partner touch (so not just down-regulating). Also greater for stranger in MFG, AG, and more for palm in pre-/post-central gyri. There is a familiarity by OT interaction in AG and MTG that seem like the same areas as those above, but for an opposing result (more for partner > stranger). MTG also increased more with OT increase in stranger first. Also, Partner v Stranger first, dorsal raphe, and hypothalamus are both involved when OT increased for partner > stranger first and decreased in stranger first (the same thematic result).I assume this does not reflect the subjective response but the up/down regulation? Or bc temporal is so involved with parsing who/what it might increase BOLD to process distinctions about who/what is happening, which has to work harder in less natural situations. Because behavioral/hormonal results are so in line with expectations, and there are so many results, and the figures are labeled as "interactions" without stating the directional info, these finer distinctions will be lost on people

Thank you very much for your very useful suggestions for improving the manuscript!

We have gone through the manuscript and added direction-of-effect information wherever possible. The most major additions or amendments we have made to the text are:

Results p 10.

“These interactions are explained by a more negative relationship between BOLD and OT change in the stranger first group, which reflects greater relative BOLD increases in individuals showing smaller percent change in OT levels. Specifically, whereas BOLD increase in SOG/cuneus showed a positive relationship in the partner first group (partner first, stranger second), these relationships were negative in the stranger first group, particularly for stranger. BOLD increase in AG showed positive or flat relationships with OT change in all conditions except for partner second in the stranger first group, which was negative (the higher the BOLD, the lower the OT change).”

…and

“For all these areas, individuals showing greater OT increase during the functional run were also more likely to show higher BOLD during partner compared to stranger touch, reflecting a more positive relationship between BOLD signal and OT change for partner (the higher the BOLD, the greater the degree of OT change). A right hemisphere cluster encompassing inferior and middle temporal gyri (ITG and MTG) also showed a main effect of OT covariate (a positive relationship between BOLD and OT change; Table 1) as well as an interaction between order and OT (Table 1; see also Figure 4A), reflecting a more positive relationship between BOLD signal and degree of OT increase in the stranger first group.

Regardless of the degree of OT change, BOLD was overall greater for stranger than partner (a main effect of familiarity) in bilateral middle frontal gyrus (MFG) and right AG, among other regions (Table 1), reflecting generally higher means but narrower range of variability for stranger than partner. BOLD was greater for palm than arm (a main effect of stimulation site; Table 1) in left postcentral gyrus (PoCG) and bilateral precentral gyrus (PrCG). No statistical interactions were observed between site of stimulation and OT covariate.”

…and

Results p 17

“Partner first vs. stranger second: For participants who had partner touch first, partner compared to stranger touch covaried with cortisol in the left ACC, right SMG, bilateral ventromedial prefrontal cortex, bilateral calcarine gyrus, right lingual gyrus, left TP and left PO (all at *p* = 0.002). These regions showed a more negative relationship between BOLD and cortisol for partner (individuals with higher BOLD had higher cortisol).”

We have also added direction information to the figure captions.

In the discussion, we expanded and elaborated on the discussion of the direction of effects in parietotemporal regions (p 22):

“Co-modulation between OT changes and BOLD in parietotemporal pathways may reflect updating of contextual information, possibly enhancing the salience of incoming sensory signals (Johnson et al., 2017; Shamay-Tsoory & Abu-Akel, 2016; Sripada et al., 2013) or of personally-relevant stimuli (Alaerts et al., 2021) following the initial encounter. […] This may reflect an enlistment of parietotemporal regions in gain control mechanisms (Grinevich & Ludwig, 2021)—more akin to a dimmer switch than an on-off button— with the “dimmer” tuning from a wider, more flexible response range under higher contextual certainty (partner first) to a narrower, less flexible range under lower contextual certainty (stranger first).”

…and

Discussion p 22

“In the present study, BOLD changes in dorsal Raphe nuclei also covaried with plasma OT (Figure 2D). Specifically, when the stranger touched first, BOLD in both hypothalamus and dorsal Raphe was greater the lower the mean OT across individuals (Figures1, 3), whereas no such modulation of OT-BOLD covariance was observed for partner. This may reflect descending modulatory influence on afferent touch signals from the periphery, manifesting here as a negative relationship between BOLD and OT during stranger touch (the BOLD signal cannot distinguish hemodynamic activity resulting from excitation or inhibition).”

The discussion does say at the top of p19, "noteworthy that the relationship…was an inverse one". Does "more stable OT" mean decreasing? or lower lows and higher highs?

We wonder the same thing, and have speculative ideas which we have now added to the Future Directions section of the discussion (page 27):

“Most human studies manipulating OT (usually via nasal administration) have been performed in male populations. In contrast, the present study used a “female-first” strategy which moves to redress this imbalance (*55*); likewise recent research in rodents has focused on OT-touch mechanisms in female samples (*1, 28*). Here we found no effect of cycle phase on evoked OT changes. An important question for future research is whether, and to what extent, these results in human females generalize to males, especially with regard to any familiarity-dependent bias in endogenous OT. Another potential sex difference may lie in the relationships between OT, cortisol, and their covariation in mPFC.”

Was this pattern similar to the ones in the citations where the directional info is lacking? I think a little more detail in this section would help.

It's difficult to compare directional influences across these studies since the manipulations were so different among them, but in general they seem to point to some degree of situational dependence of OT-related activity or behavior (and mostly more negative than positive modulation). We interpret this though the lens of OT’s likely role in attachment and nurturance of young, in which caregiving and defense can be two sides of the same coin, even though they might motivate behaviors that seem opposite (eg, nuturance vs defensive aggression).

There are still places without directional terms, especially in the discussion. E.g., end of p 15, it says "covaried with cortisol", end of p 18 "mutual influences", "affected by", "Implicated in". 2nd to last para on 19 "dorsal raphe nuclei also covaried with plasma OT", "interactions between OT and familiarity" in AG near the top of p 20. End of p23 "changes" "influenced" "differential".I get this explains the most general property if gain control produces both directions. But any time a specific result is referred to the direction should be there and then the generality at the beginning and end about how it can produce both decreases and increases will be more clear to people.Miscellaneous:Stats reporting has inconsistencies. I would italicize all stats letters and put spacing around = >< signs. If you prefer no spacing (I don't see journal formatting rules), then just make them consistent.Random errors: E.g., Burbach JP (no initials), says less rather than fewer toward the end of the discussion somewhere, there's an unattached emdash at the beginning of the discussion, and extra space after Tang et al. cite.I would prefer some NS stats for the linear mixed model with touch and core (e.g., ps >.11).I don't know if the journal has a max table/figure rule that caused these things below:Are none of the tables in the main body of the paper? The paper lists them as 1, 2, 3, like they are regular in-text tables but they were all provided as supplements (so should say S2, S3, etc?). I would prefer as a reader to have at least a few key BOLD tables in the paper. The table text could be simplified if you just number the first column (since it already says cluster at the top) and organize by area first then strength to leave all rows blank that repeat the same area. You don't have to do this, of course, but if you put a table in the main text but do not want it to take a lot of room, it would be easy to put a few simplified ones in there. Also, the T is for the t-test? It doesn't clarify. If it is a small stats t some of them would be negative right? Maybe it just refers to the peak voxel, but then IDK what the value is.The order of the figures presented in the text was confusing. 4B was never mentioned in the text? 2 and 4 are intermixed and not always in order. Can you split up the figures if needed so that only the parts relevant to the text in that place, in order, are present? There is an interesting amygdala graph in Figure 4B, but I didn't see it referred to in the text until the discussion.

We added text to point to this. Figure 4 panels were rearranged and this is now 4D.

p 19

“While BOLD signal change in left amygdala was high across conditions, it was also sensitive to partner-stranger differences (main effect of familiarity, *F*
_(1,16)_ = 5.8, *p* = 0.02), greater for the stranger in the first encounter.”

p 24

“…with left amygdala selective for stranger touch, particularly in the first encounter (Figure 4D).”

We have also gone through and made the italicization/spacing around stats signs consistent, and corrected the inclusion of author initials in the in-text ref citations, as well as the other errors you kindly brought to our attention.